# When More Data Doesn't Help: Limits of Adaptation in Multitask Learning

**Steve Hanneke** [1]   **Mingyue Xu** [1]

## Abstract

Multitask learning and related frameworks have achieved tremendous success in modern applications. In multitask learning problem, we are given a set of heterogeneous datasets collected from related source tasks and hope to enhance the performance above what we could hope to achieve by solving each of them individually. The recent work of Hanneke & Kpotufe (2022) has showed that, without access to distributional information, no algorithm based on aggregating samples alone can guarantee optimal risk as long as the sample size per task is bounded.

In this paper, we focus on understanding the statistical limits of multitask learning. We go beyond the no-free-lunch theorem in Hanneke & Kpotufe (2022) by establishing a stronger impossibility result of adaptation that holds for arbitrarily large sample size per task. This improvement conveys an important message that the hardness of multitask learning cannot be overcame by having abundant data per task. We also discuss the notion of optimal adaptivity that may be of future interests.

## 1. Introduction

Multitask learning and related concepts such as multi-source domain adaptation (Ben-David et al., 2006; 2010), transfer learning (Kuzborskij & Orabona, 2013; Cai & Wei, 2021), and meta learning/learning-to-learn (Maurer et al., 2016; Finn et al., 2017; Collins et al., 2021), to name a few, have become increasingly popular in modern applications of science and engineering, especially when data availability is an issue. Possibly, data generated by multiple related sources could be aggregated towards solving multiple learning tasks more efficiently than we could hope to solve each of them on their own. Such examples in reality include drug dis-

covery where predicting a molecule's efficacy and safety often benefits from jointly learning across many related assays and targets, robotics multi-skill learning where training perception and control jointly for several manipulation or navigation skills can accelerate learning new tasks, etc.

In this paper, we study the problem of multitask learning from a statistical perspective, where independent datasets $Z_t \sim P_t, t = 1, 2, \ldots$ are collected from multiple sources to be used for improving performance over learning individually. As a common belief, a single aggregation of datasets could benefit all tasks simultaneously. However, leveraging those noisy datasets might hurt the performance on target (any single such task). This motivates the notion of "adaptivity", asking for algorithms that can automatically identify good datasets, without possessing additional distributional information. The recent work of Hanneke & Kpotufe (2022) studied the hardness of multitask learning in terms of adaptivity. They first provided minimax upper and lower bounds for multitask learning rates accounting for the number of tasks, sample size per task, as well as certain notions of discrepancy between distributions. Their main result then revealed that adaptivity is impossible in general, namely, no learning procedure can achieve minimax rates as long as the sample size per task is constant and the learner has no additional information apart from the given multi-source data (see Subsection 3.1 for a detailed discussion).

Our work is built on top of theirs. Specifically, we address an open question remained in their paper, that is, whether having more samples per task could perhaps yield some quantifiable advantages from having multiple sources (e.g., by comparing and ranking them). To make it clear, their negative results followed from a probabilistic construction of multitasks where sample size per task needs to be bounded in terms of the model parameters. Therefore, whether adaptivity is possible when having a larger number of samples per task remains open. Our work answers negatively to this conjecture by showing a stronger impossibility result for adaptation that holds for arbitrarily large sample size per task. Notably, this improvement is substantial as it conveys an important message that the failure of adaptivity in multitask learning cannot be overcame by collecting abundant data per source task. This is because, having enough randomness in multitask distributions and sufficiently many tasks makes it hard to identify the optimal aggregation of

[1]Department of Computer Science, Purdue University, West Lafayette, IN, USA. Correspondence to: Mingyue Xu <xu1864@purdue.edu>.

*Proceedings of the 43rd International Conference on Machine Learning*, Seoul, South Korea. PMLR 306, 2026. Copyright 2026 by the author(s).

datasets, no matter how many samples can be used for comparison. Our theory reveals the following practical implication: for scientists who hope to design (uniformly) better multitask learning algorithms, they need to somehow leverage additional information rather than merely rely on the collected data. For instance, knowing which datasets are less contaminated or which sources are more relevant to the target environment would be helpful. Our proof is technical, involving an application of the Fano's method together with a tight upper bound on the KL divergence between mixture distributions. Furthermore, our results raise a fundamental problem of optimal adaptivity, i.e., what is the optimal adaptive rates (optimal rates achieved by algorithms with only access to multisource datasets) for multitask learning and what adaptive algorithms can achieve such optimal adaptive rates. We leave it as an interesting open problem for future works.

## 1.1. Related Works

Theory of multitask learning, dating back to Baxter (2000), has been extensively explored in the past. Below, we provide a brief overview of existing theoretical works on multitask learning and related areas. Crammer et al. (2008); Ben-David & Borbely (2008); Ben-David et al. (2006; 2010) built theoretical works on multitask learning based on distributional measures such as $d_A$-divergence and the $\mathcal{Y}$-discrepancy. Mansour et al. (2009) investigated the optimal data aggregation in multitask learning using a notion of non-metric discrepancy called the Rényi divergence. Ideally speaking, aggregation of multitask datasets could possibly benefit all tasks simultaneously. This has motivated a line of works (Maurer et al., 2013; Pontil & Maurer, 2013; Maurer et al., 2016) aiming to provide guarantees on average risk across tasks. For the adversarial robustness of multitask learning, the works of Qiao (2018); Qiao & Valiant (2018) have derived positive and negative results under various adversarial corruption in datasets. For multitask PAC learning with adversarial corruptions, Konstantinov et al. (2020) showed that there is a learning algorithm that overcomes the effect of corruptions and approaches the optimal test error as long as less than half of the sources are manipulated. Another bulk of theoretical works focused on the topic of multitask meta/representation learning where tasks are related by a shared latent representation/subspace (Muandet et al., 2013; Balcan et al., 2015; Tripuraneni et al., 2020; Du et al., 2021; Tripuraneni et al., 2021; Duchi et al., 2022; Collins et al., 2022; Bairaktari et al., 2023; Aliakbarpour et al., 2024; Yuan et al., 2025). Such settings are often formalized with particular structural assumptions on the shared substructures (e.g., a low-dimensional linear mapping), which are not the focus of the present work. Apart from what has already been discussed, there are other relevant literature includes online multitask learning (Balcan

et al., 2019; Denevi et al., 2019), federated/distributed learning (Yin et al., 2018; Zhu et al., 2023), and distance-based multitask learning (Duan & Wang, 2023; Gu et al., 2025).

## 1.2. Organization

The remaining of the paper is organized as follows. In Section 2, we formally describe the problem setup for multitask learning and introduce necessary definitions. In Section 3, we review the results of the previous work and outline our contributions relatively. In Section 4, we study the simple setting of agnostic multitask learning and show that adaptivity is impossible even when having unlimited data per task. In Section 5, we present our main results. Our main contribution is an extended impossibility result for adaptation under a general label noise assumption called the Bernstein class condition. We also include a detailed technical overview of the proof. In Section 6, we conclude the paper and propose some interesting future research directions. All proofs are deferred to the appendix.

## 2. Preliminaries

We consider a binary classification setting. Let $\mathcal{X}$ be an instance space and $\mathcal{Y} = \{0, 1\}$. Let $\mathcal{H} \subseteq \mathcal{Y}^{\mathcal{X}}$ be a concept class with bounded VC dimension ($\text{VC}(\mathcal{H}) = d_{\mathcal{H}} < \infty$), which we consider fixed in all subsequent discussions. Let $P$ be a data distribution on $\mathcal{X} \times \mathcal{Y}$, let $S_n = \{(x_i, y_i)\}_{i=1}^n \in (\mathcal{X} \times \mathcal{Y})^n$ be a dataset. For any concept $h \in \mathcal{H}$, define its error rate as $\text{er}_P(h) = P\{(x, y) : h(x) \neq y\}$ and its empirical error rate (on $S_n$) as $\hat{\text{er}}_{S_n}(h) = \frac{1}{n} \sum_{(x_i, y_i) \in S_n} \mathbb{1}\{h(x_i) \neq y_i\}$. For any pair $h, h' \in \mathcal{H}$, we define the excess risk of $h$ over $h'$ as $\mathcal{E}_P(h, h') = \text{er}_P(h) - \text{er}_P(h')$. Moreover, the excess risk of $h$ (over the best in class) is defined as $\mathcal{E}_P(h) = \text{er}_P(h) - \inf_{h' \in \mathcal{H}} \text{er}_P(h')$. Define the empirical excess risk of $h$ over $h'$ (on $S_n$) as $\hat{\mathcal{E}}_{S_n}(h, h') = \hat{\text{er}}_{S_n}(h) - \hat{\text{er}}_{S_n}(h')$. Finally, the $L_1(P_{\mathcal{X}})$ pseudo-distance between $h$ and $h'$ is defined as $P(h \neq h') = P_{\mathcal{X}}\{x \in \mathcal{X} : h(x) \neq h'(x)\}$, and also the empirical pseudo-distance between $h$ and $h'$ (on $S_n$) is defined as $\hat{P}_{S_n}(h \neq h') = \frac{1}{n} \sum_{(x_i, y_i) \in S_n} \mathbb{1}\{h(x_i) \neq h'(x_i)\}$. For notational simplicity, we denote an *Empirical Risk Minimization (ERM) learner* on $S_n$ by its output $\hat{h}_{S_n} = \arg\min_{h \in \mathcal{H}} \hat{\text{er}}_{S_n}(h)$.

The Bernstein class condition (Bartlett et al., 2004), is a well-known relaxation of the Tsybakov noise condition (Mammen & Tsybakov, 1999; Tsybakov, 2004), about which there is a substantial literature (e.g. Bartlett et al., 2006; Bartlett & Mendelson, 2006; Massart & Nedelec, 2006; Koltchinskii, 2006; Hanneke, 2011; Van Erven et al., 2015; Hanneke & Kpotufe, 2022). It captures a continuum from easy to hard classification, namely, the best achievable excess risk with sample size $n$ can be shown to be of order $n^{-1/(2-\beta)}$, controlled by $\beta \in [0, 1]$, which interpolates between $n^{-1}$ and

$n^{-1/2}$. It is formally defined as follow.

**Definition 2.1** (**Bernstein class condition**). Let $\beta \in [0, 1]$ and $C_\beta \geq 2$. We say that a distribution $P$ satisfies a *Bernstein class condition* with parameters $(C_\beta, \beta)$ if

$$P(h \neq h_P^*) \leq C_\beta \cdot (\mathcal{E}_P(h))^\beta, \quad \forall h \in \mathcal{H},$$

where $h_P^*$ satisfies $h_P^* \in \arg\min_{h \in \mathcal{H}} \mathrm{er}_P(h)$.

Note that the above Bernstein class condition always holds when $\beta = 0$ since $P(h \neq h_P^*) \leq 1$. Moreover, $P(h \neq h_P^*) \geq \mathcal{E}_P(h)$ always holds. It is worth mentioning that, when the Bayes-optimal classifier is assumed in class, the Tsybakov noise condition implies the Bernstein class condition. Also, the classic Massart's noise condition implies a linear Bernstein class condition, i.e., $\beta = 1$. Our next definition extends the intuition that how well one can do in a multitask learning problem depend on how closely two distributions relate to each other. We introduce the notion of *transfer exponent*, which is commonly adopted in the literature of transfer learning and multitask learning (Hanneke & Kpotufe, 2019; 2022; Hanneke et al., 2023; Hanneke & Kpotufe, 2024; Kalan et al., 2025).

**Definition 2.2** (**Transfer exponent**). Let $\rho > 0$ and $C_\rho \geq 2$. We say that a distribution $P$ has a *transfer exponent* $\rho$ to a distribution $Q$ with respect to $\mathcal{H}$ if

$$\mathcal{E}_Q(h) \leq C_\rho \cdot (\mathcal{E}_P(h))^{1/\rho}, \quad \forall h \in \mathcal{H}.$$

Note that for any pair of distributions $(P, Q)$, $P$ always has a transfer exponent $\rho = \infty$ to $Q$. Indeed, it has been shown that the transfer exponent tightly captures the minimax rates of transfer learning (Hanneke & Kpotufe, 2019) as well as the minimax rates of multitask learning (Hanneke & Kpotufe, 2022). Concretely, a transfer exponent $\rho$ reveals a size $n_P^{1/\rho}$ data contributes into the transfer learning from $P$ to $Q$. Accordingly, $\rho \to \infty$ implies the growing difficulty of transfer, and $\rho < 1$ implies that source data could be more useful than target data for learning, which is called super-transfer.

With those aforementioned definitions, we are now able to formalize the problem of multitask learning. We consider a setting where there are $N$ source environments associated with source distributions $\{P_t\}_{t=1}^N$ and a target environment associated with a target distribution $\mathcal{D}$. For each $1 \leq t \leq N$, we have a dataset $Z_t$ i.i.d. from $P_t$ with $|Z_t| = n_t$. For the target $\mathcal{D}$, we have an i.i.d. dataset $Z_\mathcal{D}$ with $|Z_\mathcal{D}| = n_\mathcal{D}$. The goal is to aggregate multiple source datasets $\{Z_t\}_{t=1}^N$ towards improving the learning performance on target $\mathcal{D}$.

Moreover, we make the following assumptions:

(1) All data distributions induce the same optimal classifier, i.e., $\exists h^* \in \mathcal{H}$ such that $h^* \in \arg\min_{h \in \mathcal{H}} \mathcal{E}_P(h)$, $\forall P \in \{P_1, \ldots, P_N, \mathcal{D}\}$.

(2) For each $1 \leq t \leq N$, source $P_t$ has a transfer exponent $\rho_t$ to the target distribution $\mathcal{D}$ with respect to $\mathcal{H}$, with a constant $C_\rho \geq 2$. Note that $\mathcal{D}$ always has a transfer exponent $\rho_\mathcal{D} = 1$ to itself.

(3) All sources $P_t, \forall 1 \leq t \leq N$ and target $\mathcal{D}$ satisfy a Bernstein class condition with parameters $(C_\beta, \beta)$.

We would like to emphasize that, the stronger the assumptions are, the stronger a negative result (lower bound) is. Taking the assumption (1) as an instance, our no-free-lunch theorem implies that adaptation is impossible even if all multitask distributions share the same optimal classifier. Hence, adaptation is certainly impossible when $h^*$ differs across tasks. On the other hand, as discussed earlier, with additional structural assumptions, (1) is usually relaxed to an assumption that multiple models share a common feature which is called representation (Ando et al., 2005; Argyriou et al., 2008; Kumar & Daumé III, 2012; Du et al., 2021; Tripuraneni et al., 2021). Since the main focus of this work is to develop a fundamental guarantee on statistical learning, we make a more general assumption as adopted in Hanneke & Kpotufe (2022) that all tasks share the same optimal classifier, not just a common representation. This scenario is also of independent interest in other areas, e.g., invariant risk minimization (Arjovsky et al., 2019). For notational simplicity, we can write $P_{N+1} = \mathcal{D}$, $n_{N+1} = n_\mathcal{D}$, $Z_{N+1} = Z_\mathcal{D}$ and $\rho_{N+1} = \rho_\mathcal{D} = 1$. Throughout the paper, we will be interested in the order statistics of transfer exponents defined in (2), i.e., $\rho_{(1)} \leq \rho_{(2)} \leq \cdots \leq \rho_{(N+1)}$. Let $P_{(t)}, Z_{(t)}, n_{(t)}$ denote the corresponding distribution, dataset and sample size indexed by $(t)$ in order. We define the *averaged transfer exponent* as $\bar{\rho}_t = \sum_{s \in [t]} n_{(s)} \rho_{(s)} / \sum_{s \in [t]} n_{(s)}$ for any $1 \leq t \leq (N+1)$. Let $\mathcal{M} = \mathcal{M}(C_\beta, \beta, C_\rho, \{\rho_t\}_{t \in [N+1]}, \{Z_t\}_{t \in [N+1]})$ represent the class of multitasks that satisfies (1) – (3). Proven by Hanneke & Kpotufe (2022), the minimax rates of multitask learning are

$$\inf_{\hat{h}} \sup_{\Pi \in \mathcal{M}} \mathbb{E}_\Pi \left[ \mathcal{E}_\mathcal{D}(\hat{h}) \right] \asymp \min_{t \in [N+1]} \left( \sum_{s \in [t]} n_{(s)} \right)^{-1/(2-\beta)\bar{\rho}_t},$$

where $\hat{h}$ is any multitask learner possessing the knowledge of $\mathcal{M}$ and data $Z \sim \Pi$.

## 3. Main Results

Consider a simple scenario where a concept class contains only two hypotheses $\mathcal{H} = \{h_0, h_1\}$ mapping from $\mathcal{X} = \{x_0, x_1\}$ to $\mathcal{Y} = \{0, 1\}$. Assume that $h_0(x_0) = h_1(x_0)$, $h_0(x_1) = 0$ and $h_1(x_1) = 1$. Also, assume that the shared target concept $h^* \in \mathcal{H}$ satisfies $h^*(x_1) = y^* \in \{0, 1\}$ and denote the remaining concept by $h \in \mathcal{H} \setminus \{h^*\}$, i.e., $\mathcal{H} = \{h^*, h\}$. The following lemma interpret the Bernstein class condition (Definition 2.1) under this simple setting.

**Lemma 3.1.** *Let $P$ be any distribution supported on $\mathcal{X} \times \mathcal{Y}$ such that $h^* = \arg\min_{h \in \mathcal{H}} er_P(h)$. Then, $P$ satisfies the Bernstein class condition (Definition 2.1) if and only if*

$$P(Y = y^*|X = x_1) \geq \frac{1}{2}\left(1 + C_\beta^{-1/\beta}\left[P_X(x_1)\right]^{1/\beta - 1}\right).$$

### 3.1. Background

Before we proceed to the statement of our main results, we elucidate the motivation of this work. To this end, we start by reviewing the recent work of Hanneke & Kpotufe (2022) which provides a no-free-lunch type theorem for multitask learning, namely, besides certain regimes of adaptivity (e.g., $\beta = 1$), no adaptive algorithm can achieve the minimax rates without possessing distributional information. To formalize, an adaptive algorithm is a reasonable procedure that only has access to a multisample $Z \sim \Pi$ for some unknown $\Pi \in \mathcal{M}$, but no access to any prior information on (the parameters) of $\mathcal{M}$, e.g., the correspondence between $\{Z_t\}_{t \in [N+1]}$ and $\{\rho_t\}_{t \in [N+1]}$. To prove their negative result, they utilized the following multitask construction. Let $\sigma \in \{\pm 1\}$. Assume that the optimal classifier $h^*$ predicts 1 on $x_0$ and $\sigma$ on $x_1$. (For binary classification, we can consider equivalently $\mathcal{Y} = \{\pm 1\}$ instead of $\mathcal{Y} = \{0, 1\}$.) Now, it is clear that learning $h^*$ is equivalent to learning the true label $\sigma$. They considered the following three types of distributions:

- **Target:** $\mathcal{D}_\sigma = \mathcal{D}_X \times \mathcal{D}_{Y|X}^\sigma$ where $\mathcal{D}_X(x_1) = (1/2)\epsilon_0^\beta$, $\mathcal{D}_X(x_0) = 1 - (1/2)\epsilon_0^\beta$, $\mathcal{D}_{Y|X}^\sigma(Y = 1|X = x_1) = 1/2 + \sigma c_0 \epsilon_0^{1-\beta}$ and $\mathcal{D}_{Y|X}^\sigma(Y = 1|X = x_0) = 1$ with some constant $c_0 > 0$.
- **Benign Source:** $P_\sigma = P_X \times P_{Y|X}^\sigma$ where $P_X(x_1) = 1$ and $P_{Y|X}^\sigma(Y = 1|X = x_1) = 1/2 + \sigma/2$.
- **Noisy Source:** $Q_\sigma = Q_X \times Q_{Y|X}^\sigma$ where $Q_X(x_1) = c_1\epsilon^\beta$, $Q_X(x_0) = 1 - c_1\epsilon^\beta$, $Q_{Y|X}^\sigma(Y = 1|X = x_1) = 1/2 + \sigma\epsilon^{1-\beta}$ and $Q_{Y|X}^\sigma(Y = 1|X = x_0) = 1$ with some constant $c_1 > 0$.

They assumed the following: each source dataset is of size $n$, i.e., $|Z_t| = n$ for every $t \in [N]$. There are $N_P$ benign sources ($Z_t \sim P_\sigma$) and $N_Q$ noisy sources ($Z_t \sim Q_\sigma$). Moreover, we set $\epsilon_0 = 1 \wedge n_{\mathcal{D}}^{-1/(2-\beta)}$ and $\epsilon = (n \cdot N_P)^{-1/(2-\beta)}$. By Lemma 3.1, it is not hard to check that $\mathcal{D}_\sigma$, $P_\sigma$ and $Q_\sigma$ satisfy the Bernstein class condition with parameters $\beta$ and $C_\beta = \max\{(1/2)c_0^{-\beta}, 2\}$. The no-free-lunch theorem proven by Hanneke & Kpotufe (2022) can be described as follow: for any adaptive learner $\hat{h}$, there exists a multitask model $\Pi \in \mathcal{M}$ such that $\mathbb{E}_\Pi[\mathcal{E}_{\mathcal{D}}(\hat{h})] = \Omega(n_{\tilde{\mathcal{D}}}^{-1/(2-\beta)})$. However, there exists a semi-adaptive learner $\tilde{h}$ such that $\sup_{\Pi \in \mathcal{M}} \mathbb{E}_\Pi[\mathcal{E}_{\mathcal{D}}(\tilde{h})] = O((n \cdot N_P)^{-1/(2-\beta)})$.[1]

We underline that their negative result was proven under two

---

[1]It is a learner that has access to the ranking information of the transfer exponents.

constraints: $n < 2/\beta - 1$ and $N = \Omega(\exp(n))$. Indeed, for their construction, the following requirements are necessary:

$$N_Q \geq 3N_P \quad \text{and} \quad N_Q^{\frac{2-(n+1)\beta}{2-\beta}} \geq 2^{15n} \cdot N_P^2.$$

Note that the above implicitly requires $n < 2/\beta - 1$. It is actually a strict constraint since for a large $\beta \in (0, 1)$, this negative result is essentially vacuous. Some intuition on where these requirements arise can be derived from their proof technique. From the benign sources $P_\sigma$, one gets perfectly only samples $(x_1, \sigma)$. However, from those noisy sources $Q_\sigma$, one may obtain samples of homogeneous vectors but with identical flipped labels $(x_1, -\sigma)$, making it hard to guess which sources are the ground-truth when $N_Q \gg N_P$. Specifically, they need the following:

1. $N_Q \gg N_P$, so that pooling is suboptimal.
2. $n < 2/\beta - 1$, so that the learner cannot tell whether a single source is benign or noisy (by comparing homogeneous vectors), based on the dataset drawn from its underlying distribution.
3. $N_Q = \Omega(\exp(n))$ and thus $N = \Omega(\exp(n))$, so that even having access to all (source and target) datasets, the learner cannot tell whether the correct label is $\sigma$ or $-\sigma$ by comparing the numbers of homogeneous vectors $(x_1, \sigma)$ and $(x_1, -\sigma)$.

### 3.2. Our New Setting

One of the main results of this work is a stronger impossibility result for adaptation in multitask learning. Specifically, while still requiring $N = \Omega(\exp(n))$, we prove that a negative result holds with an arbitrarily large sample size $n$. In the following, let us briefly explain how we get rid of the constraint $n < 2/\beta - 1$. Let $\beta \in (0, 1)$. Assume that the optimal function $h^* \in \mathcal{H}$ predicts 1 on $x_0$ and $y^* \in \{0, 1\}$ on $x_1$. Let $\epsilon, \epsilon_0 \in (0, 1)$ such that $\epsilon_0 < \epsilon$ (which will be specified later). Our new multitask setting is constructed based on the following two types of distributions:

- **Fair Source:** $P = P_X \times P_{Y|X}$ where $P_X(x_1) = C_\beta \epsilon^\beta$, $P_X(x_0) = 1 - C_\beta \epsilon^\beta$, $P_{Y|X}(Y = y^*|X = x_1) = 1/2 + (1/2)C_\beta^{-1}\epsilon^{1-\beta}$ and $P_{Y|X}(Y = 1|X = x_0) = 1$.
- **Noisy Source:** $Q = Q_X \times Q_{Y|X}$ where $Q_X(x_1) = C_\beta \epsilon_0^\beta$, $Q_X(x_0) = 1 - C_\beta \epsilon_0^\beta$, $Q_{Y|X}(Y = y^*|X = x_1) = 1/2 + (1/2)C_\beta^{-1}\epsilon_0^{1-\beta}$, and $Q_{Y|X}(Y = 1|X = x_0) = 1$.

$Q$ is called a noisy distribution (noisier than $P$) because $\epsilon_0 < \epsilon$. First, it is not hard to check the Bernstein class condition for these distributions via Lemma 3.1. For distribution $P$, we have $2P(Y = y^*|X = x_1) - 1 = C_\beta^{-1}\epsilon^{1-\beta} \geq C_\beta^{-1/\beta}[P_X(x_1)]^{1/\beta - 1}$. Similarly for $Q$, we have $2Q(Y = y^*|X = x_1) - 1 = C_\beta^{-1}\epsilon_0^{1-\beta} \geq C_\beta^{-1/\beta}[Q_X(x_1)]^{1/\beta - 1}$.

Furthermore, we calculate $\mathcal{E}_P(h) = (2P(Y = y^*|X = x_1) - 1)P_X(x_1) = \epsilon$ and $\mathcal{E}_Q(h) = (2Q(Y = y^*|X = x_1) - 1)Q_X(x_1) = \epsilon_0$.

Assume that each source distribution in $\{P_{(t)}\}_{t \in [N]}$ is either $P$ or $Q$, and the target distribution is simply $\mathcal{D} = P$. Then, it is clear that $P$ has a transfer exponent $\rho_P = 1$ to $\mathcal{D}$ and $Q$ has a transfer exponent $\rho_Q > 1$. Suppose that we have a size-$n$ dataset from each source, but do not have any data from the target. Note that making this assumption does not lose any generality since $\mathcal{D} = P$. Let $1 \le t^* \le N$ and assume that we have $t^*$ sources $P$ and $(N - t^*)$ sources $Q$. In other words, by ranking the transfer exponents, we have $P_{(t)} = P$ for every $1 \le t \le t^*$ and $P_{(t)} = Q$ for every $t^* < t \le N$. Finally, we set $\epsilon = (n\sqrt{N})^{-1/(2-\beta)}$, $\epsilon_0 = (nN)^{-1/(2-\beta)}$ and $t^* = \sqrt{n^{n\beta/(2-\beta)}N}$.

The intuition behind is that, here, both two types of source distributions are very noisy, unlike the case in Subsection 3.1 where the benign distribution is almost perfect. Therefore, given an arbitrarily large sample size $n$, there is a sufficiently large $N = \Omega(n^{n\beta/(1-\beta)})$ such that one cannot tell whether a source is fair or noisy. Indeed, $t^*$ is assumed to be the optimal (minimax) cut-off index which yields the minimax rates $O((nt^*)^{-1/(2-\beta)})$. However, given sufficiently many tasks (a sufficiently large $N$), we show that no adaptive algorithm can achieve a rate better than $\Omega(\epsilon) = \Omega((n\sqrt{N})^{-1/(2-\beta)})$. Since $(nt^*)^{-1/(2-\beta)} = o((n\sqrt{N})^{-1/(2-\beta)})$, this yields a stronger no-free-lunch theorem.

We emphasize that, though sharing a similar high-level idea to the previous work, our proof strategy is different to theirs. Besides having a different hard-case construction, our proof technique is to directly bound the KL-divergence between the likelihood functions (of mixture distributions) when data are generated according to opposite labels. In contrast, the lower bound proof in Hanneke & Kpotufe (2022) relies on the fact that likelihood-ratio test is optimal. Note that bounding likelihood-ratio is conceptually equivalent to bounding the KL-divergence in general.[2] However, as discussed earlier, their hard-case construction is relatively strong, containing certain amount of benign tasks. And it is indeed this strong setup allows them to lower bound the likelihood-ratio by a decomposition into the contributions of homogeneous vs non-homogeneous examples (see their Proposition 2). This decomposition significantly reduces the technical difficulty. Basically, it reduces the problem from bounding a KL-divergence between mixtures to bounding a KL-divergence between joint distributions with pairwise independent marginals. In other words, their strong construction facilitates the analysis, but their lower bound pays the price of having a significant constraint of $n < 2/\beta - 1$.

[2]Bounding above the KL-divergence is equivalent to bounding below a likelihood-ratio statistic.

Our proof consists of an application of the Fano's method together with a tight upper bound on the KL divergence between mixture of distributions, providing an information-theoretic basis for hypothesis testing. It is worth mentioning that, bounding the KL divergence between mixtures is in general very challenging (Hershey & Olsen, 2007). There are no known systematic guarantees better than leveraging Jensen's inequality. Finally, Hanneke & Kpotufe (2022) has showed that pooling is nearly (minimax) optimal whenever $\beta = 1$, but is in general suboptimal for $\beta \in (0, 1)$. It is then natural to ask whether pooling is always an optimal adaptive algorithm, that is, optimal among all adaptive algorithms. In this paper, we reject this conjecture by providing counter-examples. It could be an interesting future direction to understand what is the optimal adaptive algorithm for multitask learning.

## 4. The Agnostic Case

Before proceeding to our main results, we take an additional step of studying the classical agnostic setting, i.e. $\beta = 0$ in this section. Note that the agnostic case puts no assumption on the label noise and is thus more general (weaker) than the Bernstein class condition. We show that, adaptation is impossible in the agnostic case even if we allow unlimited data per task. While this lower bound does not guarantee whether we can remove the constraint $n < 2/\beta - 1$ since $\beta = 0$, it helps to develop some intuitions on what makes our construction in Subsection 3.2 different from the one in Subsection 3.1, and sheds light on proving such a lower bound under the stronger Bernstein class condition.

We consider the following two types of distributions: for any $\delta \in (0, 1)$ and $n \in \mathbb{N}$, let $\epsilon = \epsilon(\delta, n) = \sqrt{(1/n)\log(1/\delta)}$ and

- $P = P_X \times P_{Y|X}$ where $P_X(x_1) = 1$ and $P_{Y|X}(Y = y^*|X = x_1) = 1/2 + \epsilon$.
- $Q = Q_X \times Q_{Y|X}$ where $Q_X(x_1) = 1$ and $Q_{Y|X}(Y = y^*|X = x_1) = 1/2$.

Assume that we have an i.i.d. dataset of size $n$ from each of the $N$ source tasks, where only one of them is associated with distribution $P$ and the remaining $(N - 1)$ of them are associated with distribution $Q$. Assume that the target distribution $\mathcal{D} = P$ and there is no target data. Clearly, the optimal classifier $h^* \in \mathcal{H}$ predicts $y^* \in \{0, 1\}$ on $x_1$. The following theorem implies that given additional information on distributions and the corresponding datasets, there is an algorithm that achieves optimal excess risk.

**Theorem 4.1.** *Let $Z_P \sim P$ with $|Z_P| = n$ be the dataset from the source associated with distribution $P$. Let $\hat{h}_{Z_P} = \mathrm{ERM}(Z_P)$ denote the output of ERM over $Z_P$. We have $\mathbb{P}(\hat{h}_{Z_P} \ne h^*) \le \delta$.*

Since $\mathcal{E}_{\mathcal{D}}(h) = 2\epsilon$, it implies that $\mathbb{E}[\mathcal{E}_{\mathcal{D}}(\hat{h}_{Z_P})] \leq 2\delta\epsilon$. Next, we have the following complementary lower bound. It basically states that, without having knowledge of which source task/dataset is good (which dataset is from $P$), no adaptive algorithm can guarantee a non-trivial risk.

**Theorem 4.2.** *Let $Z_t$ be an i.i.d. dataset with $|Z_t| = n$ from each source $t \in [N]$. Let $\mathcal{A}$ be any learning algorithm that only has access to $Z = \bigcup_{t \in [N]} Z_t$, but has no knowledge of the multitasks. Let $\hat{h}_Z = \mathcal{A}(Z)$. If $N \geq \delta^{-8}$, we have $\mathbb{P}(\hat{h}_Z \neq h^*) \geq (2 - \sqrt{2})/4$.*

We point out some issues of this example. First, per aforementioned, $n < 2/\beta - 1$ is indeed vacuous when $\beta = 0$. Therefore, studying the agnostic case does not prove or disprove that a negative result can hold without such a sample size constraint. Moreover, Theorem 4.2 implies that any adaptive algorithm $\mathcal{A}$, without possessing the knowledge of multitask distributions, has $\mathbb{E}[\mathcal{E}_{\mathcal{D}}(\mathcal{A}(Z))] = \Omega(\epsilon)$. However, note that $\rho_P = 1$ and $\rho_Q = \infty$ (to the target $\mathcal{D} = P$). It is not hard to calculate that the minimax rate is also of order $\epsilon$. Therefore, this example does not rigorously exclude the possibility of adaptivity. Here, what might confuse the reader is that the algorithm in Theorem 4.1 behaves better than the minimax rates. This could happen since the minimax rates account for the worst case optimal rates, namely, the best achievable uniform rates over all multitask distributions that admit a set of transfer exponents (see Hanneke & Kpotufe (2022) for details).

Despite all the above, the results for the agnostic case reveal some interesting aspects on the hardness of adaptation. First, certain margin conditions that quantify the amount of label noise (e.g., Bernstein class condition with $\beta \in (0, 1)$) are crucial when understanding the hardness of adaptation. Additionally, note that Theorem 4.2 only needs $N \geq \delta^{-8}$. By contrast, $N = \Omega(\exp(n))$ is necessary to the construction of lower bound in both Hanneke & Kpotufe (2022) and our Section 5. This raises an interesting question that what is the minimal assumption on the number of tasks $N$ to allow adaptivity (see Section 6 for a further discussion). Last but importantly, this simple example demonstrates the suboptimality of pooling (pool all datasets together and run a global ERM) for adaptation in multitask learning, as we will show below.

Specifically, we demonstrate that a fast learning rate that is beyond pooling's capacity can be achieved by some better adaptive learner (Algorithm 1). Consider the same multitask distributions but now with $\epsilon = \Omega(1)$. Intuitively, this makes $P$ quite different from $Q$ so that identifying good datasets could be possible and pooling will be catastrophic. The next theorem provides a lower bound for the pooling rates.

**Theorem 4.3.** *Let $Z = \bigcup_{t \in [N]} Z_t$ and $\hat{h}_{pool} = \mathrm{ERM}(Z)$ denote the output of pooling. Then, if $N = \Omega(n)$, we have*

$$\mathbb{E}[\mathcal{E}_{\mathcal{D}}(\hat{h}_{pool})] = \Omega(1).$$

---

**Algorithm 1** Intersection of Bernstein Balls

**input** Multitasks $(C_\beta, \beta, C_\rho, \{\rho_t\}_{t \in [N]}, \{Z_t\}_{t \in [N]})$, concept class $\mathcal{H}$, constant $C_0$ in Lemma E.9, confidence level $\delta \in (0, 1)$.
1: Let

$$\epsilon(n, \delta) = \frac{d_{\mathcal{H}}}{n} \log\left(\frac{n}{d_{\mathcal{H}}}\right) + \frac{1}{n} \log\left(\frac{1}{\delta}\right).$$

2: Initialize $\hat{\mathcal{H}} \leftarrow \mathcal{H}$, $Z \leftarrow \emptyset$.
3: **for** $t = 1, \ldots, N$ **do**
4:  Let $\delta_t = \delta/(6t^2)$.
5:  Calculate the subset

$$\mathcal{H}_t = \Big\{ h \in \mathcal{H} : \hat{\mathcal{E}}_{Z_t}(h, \hat{h}_{Z_t}) \leq$$
$$C_0 \sqrt{\hat{P}_{Z_t}(h \neq \hat{h}_{Z_t}) \cdot \epsilon(n, \delta_t)} + C_0 \cdot \epsilon(n, \delta_t) \Big\}.$$

6: **end for**
**output** Any $h$ in $\bigcap_{t=1}^{N} \mathcal{H}_t$.

---

The following theorem shows that Algorithm 1 adapts to the designed multitask problem much better than pooling.

**Theorem 4.4** (**Upper bound of Algo. 1**). *Let $\hat{h}_{IB}$ be the output of Algorithm 1. Under the setting of Theorem 4.3, we have $\mathbb{E}[\mathcal{E}_{\mathcal{D}}(\hat{h}_{IB})] = O(1/n)$. Moreover, for any multitask distributions that admit the sequence of transfer exponents $\{1, \infty, \ldots, \infty\}$, we have $\mathbb{E}[\mathcal{E}_{\mathcal{D}}(\hat{h}_{IB})] = O(\sqrt{\log(N)/n})$. Note that the minimax rate is $O(1/\sqrt{n})$.*

Theorem 4.4 states that: (i) For the specific multitask problem constructed in this section, Algorithm 1 not only outperforms pooling, but also admits learning rates faster than the minimax rates. (ii) Algorithm 1 guarantees nearly optimal uniform rates[3] (differing only by a factor of $\log(N)$ to the minimax rates), given a set of transfer exponents $\{1, \infty, \ldots, \infty\}$. Note that the multitask distributions constructed in this section also admits $\{1, \infty, \ldots, \infty\}$ transfer exponents. Hence, we believe that Algorithm 1 might serve as a base for designing an optimal adaptive multitask learner.[4]

## 5. Impossibility of Adaptation

In this section, we formally state the main results of this paper. Specifically, we go beyond the limitation mentioned in Section 4, that is, we find a multitask setting with a non-

---

[3]The bound holds uniformly over all multitask distributions that admit a set of transfer exponents.

[4]We refer the readers to Section 6 for the concept of optimal adaptivity.

trivial $\beta \in (0,1)$ (Subsection 5.1) such that a no-free-lunch type of theorem holds with arbitrarily large sample size per task $n > 2/\beta - 1$. We will show that no adaptive algorithm can achieve the minimax rates, while some optimal algorithm that possesses additional information can (Subsection 5.2). More interestingly, we show in Subsection 5.3 that pooling achieves nearly optimal adaptive rates in this multitask problem.

### 5.1. Formalization

Recall the multitask distributions defined in Subsection 3.2:

- **Fair Source:** $P = P_X \times P_{Y|X}$ where $P_X(x_1) = C_\beta \epsilon^\beta$, $P_X(x_0) = 1 - C_\beta \epsilon^\beta$, $P_{Y|X}(Y = y^*|X = x_1) = 1/2 + (1/2)C_\beta^{-1}\epsilon^{1-\beta}$ and $P_{Y|X}(Y = 1|X = x_0) = 1$.
- **Noisy Source:** $Q = Q_X \times Q_{Y|X}$ where $Q_X(x_1) = C_\beta \epsilon_0^\beta$, $Q_X(x_0) = 1 - C_\beta \epsilon_0^\beta$, $Q_{Y|X}(Y = y^*|X = x_1) = 1/2 + (1/2)C_\beta^{-1}\epsilon_0^{1-\beta}$, and $Q_{Y|X}(Y = 1|X = x_0) = 1$.

Let $Z_t \sim P_t^n$ with $|Z_t| = n$ be an i.i.d. dataset from each source $t \in [N]$. Assume that the target $\mathcal{D} = P$ and we have no target data. Let $Z_{(1)}, \ldots, Z_{(N)}$ be ordered according to the ordered statistics of transfer exponents $\rho_{(1)} \leq \rho_{(2)} \leq \ldots \leq \rho_{(N)}$. Now, let us specify the noisy levels $\epsilon$ and $\epsilon_0$ as

$$\epsilon = \left(\frac{1}{n\sqrt{N}}\right)^{1/(2-\beta)} \quad \text{and} \quad \epsilon_0 = \left(\frac{1}{nN}\right)^{1/(2-\beta)}, \quad (1)$$

and make the following assumptions

$$N \geq n^{n\beta/(1-\beta)} \quad \text{and} \quad t^* = \sqrt{N \cdot n^{n\beta/(2-\beta)}}. \quad (2)$$

Our construction follows from the following intuitions. For any learning algorithm $\mathcal{A}$ given $Z$ as input, $\mathcal{E}_\mathcal{D}(\mathcal{A}(Z)) = \epsilon$ if it outputs incorrectly that $\mathcal{A}(Z) \neq h^*$. When the number of good datasets is $N_P = \Omega(t^*)$, the minimax rates are at least as fast as $(nt^*)^{-1/(2-\beta)}$. In other words, it serves as an upper bound on the minimax rates, which suffices for our purpose. Therefore, if we can show that $\mathcal{A}(Z) \neq h^*$ holds with some constant probability, it implies that $\mathbb{E}[\mathcal{E}_\mathcal{D}(\mathcal{A}(Z))] = \Omega(\epsilon) = \Omega((n\sqrt{N})^{-1/(2-\beta)})$. Since $(nt^*)^{-1/(2-\beta)} = o((n\sqrt{N})^{-1/(2-\beta)})$, it yields that $\mathcal{A}$ is not adaptive (see the following Subsection 5.2 for details). When $\beta = 0$, our result simply implies a lower bound of the minimax rates similar to Section 4.

Note that a (super) exponential dependence $N = \Omega(\exp(n))$ is required here, while only a constant (in terms of $n$) number of tasks $N$ is required in Theorem 4.2. This suggests the following two possibilities: (i) A negative result can even hold for arbitrarily large $n > 2/\beta - 1$ and $N = \Omega(\text{poly}(n))$ number of tasks (see Section 6 for a discussion on why $\Omega(\text{poly}(n))$ is considered); or (ii) Adaptivity could be possible when having a smaller number of tasks, for instance,

when $N = o(\exp(n))$. We leave it as an open question to answer in future.

### 5.2. A Stronger No-free-lunch Theorem

We first show that given the knowledge of the transfer exponents, there is a learning algorithm that can guarantee good learning performance. Concretely, we show in the following Theorem 5.1 that ERM over all datasets from fair sources achieves a learning rate even faster than the minimax rate. As mentioned earlier, the minimax rates concern the worst-case scenario of multitask distributions that admit a given set of transfer exponents. Therefore, for a specific multitask problem with fixed distributions, it is very likely that learning rates faster than the minimax rates are achievable. Clearly, the described algorithm is not adaptive as it is assumed to possess the knowledge of which datasets are good (from those fair sources).

**Theorem 5.1.** *Let $\epsilon$ and $\epsilon_0$ be defined as in (1). For any $n > 0$, assume that (2) holds. Let $Z^* = \bigcup_{t \in [t^*]} Z_{(t)}$ and $\hat{h}_{Z^*} = \text{ERM}(Z^*)$. Then, we have*

$$\mathbb{E}\left[\mathcal{E}_\mathcal{D}(\hat{h}_{Z^*})\right] = O\left(\left(n\sqrt{N}\right)^{-1/(2-\beta)} e^{-n^{n\beta/2(2-\beta)}}\right).$$

While still getting dominated by the term $(n\sqrt{N})^{-1/(2-\beta)}$, the above rates exceed the minimax rates as $e^{-n^{n\beta/2(2-\beta)}} = o(n^{-n\beta/2(2-\beta)^2})$. Next, we formally state our main result, a stronger no-free-lunch theorem for multitask learning that holds for arbitrarily large sample size per task $n > 2/\beta - 1$, stating that the minimax rates are not attainable by any adaptive algorithm that only has access to multisource data.

**Theorem 5.2** (**Adaptivity is Impossible**). *Let $\epsilon$ and $\epsilon_0$ be defined as in (1). For any $n > 0$, assume that (2) holds. Let $Z = \bigcup_{t \in [N]} Z_t$. Then, for any adaptive learning algorithm $\mathcal{A}$ given only $Z$ as input, we have*

$$\mathbb{E}[\mathcal{E}_\mathcal{D}(\mathcal{A}(Z))] = \Omega\left(\left(n\sqrt{N}\right)^{-1/(2-\beta)}\right).$$

Let us briefly discuss a sketch of the proof for Theorem 5.2. Details are deferred to Appendix C. We will consider a random construction of multitask setting, which makes the calculation of KL-divergence simple due to the independence between tasks and facilitate the analysis. This technique has also been adopted in Hanneke & Kpotufe (2022) where it was called an "Approximate Multitask" model.

**Notations.** Concretely, let $h^*(x_1) = y^* = \sigma \in \{0,1\}$, let $t^* \in [N]$, $\alpha_F = t^*/N$ and $\alpha_N = 1 - \alpha_F = 1 - t^*/N$. For each $t \in [N]$, the source task $t$ has a distribution $P_t = P(\sigma)$ with probability $\alpha_F$ and has a distribution $P_t = Q(\sigma)$ with probability $\alpha_N$. For notational simplicity, we abbreviate $P(\sigma)$ and $Q(\sigma)$ to $P$ and $Q$, respectively. Let $\boldsymbol{I}^* \subset [N]$

be the set of indices where $P_t = P$ for every $t \in \boldsymbol{I}^*$. We know from the construction that $|\boldsymbol{I}^*| \sim \mathrm{Bin}(N, \alpha_F)$. For every $t \in [N]$, let $\hat{n}_{x_1}(Z_t) = \hat{n}_0(Z_t) + \hat{n}_1(Z_t)$ denote the number of examples in $Z_t$ with marginals at $x_1$, where $\hat{n}_0(Z_t)$ and $\hat{n}_1(Z_t)$ denote the number of $(x_1, 0)$ and $(x_1, 1)$ examples in the dataset $Z_t$, respectively. Let $Z_{x_1}$ denote all the examples in $Z$ with marginals at $x_1$, and thus $|Z_{x_1}| = \sum_{t=1}^{N} \hat{n}_{x_1}(Z_t)$. Finally, we formalize our learning problem as a binary hypothesis testing problem described as follow:

> **Problem.** *Let $\sigma \sim \mathrm{Unif}(\{0, 1\})$. Consider two distributions $P_\sigma, \sigma \in \{0, 1\}$. We get samples $Z = \bigcup_{t=1}^{N} Z_t$ generated via the following: for every $t \in [N]$, $Z_t \sim P(\sigma)$ with probability $\alpha_F$ and $Z_t \sim Q(\sigma)$ with probability $1 - \alpha_F = \alpha_N$, where*
> - *$P(\sigma)$ satisfies $P_X(x_1) = C_\beta \epsilon^\beta$, $P_{Y|X}(Y = \sigma | X = x_1) = 1/2 + (1/2)C_\beta^{-1}\epsilon^{1-\beta}$ and $P_{Y|X}(Y = 1 | X = x_0) = 1$.*
> - *$Q(\sigma)$ satisfies $Q_X(x_1) = C_\beta \epsilon_0^\beta$, $Q_{Y|X}(Y = \sigma | X = x_1) = 1/2 + (1/2)C_\beta^{-1}\epsilon_0^{1-\beta}$ and $Q_{Y|X}(Y = 1 | X = x_0) = 1$.*
>
> *Our goal is to distinguish probability distributions $P_0$ from $P_1$ given samples $Z$. Let the null hypothesis $H_0$ claims that $Z$ comes from $P_\sigma$, and the alternative hypothesis $H_1$ claims $Z$ from $P_{1-\sigma}$. We define a **test** as a function $f : \mathcal{Z} \mapsto \{0, 1\}$, that given samples $Z$, indicates which hypothesis should be true: $f(Z) = \sigma \Rightarrow H_0$ and $f(Z) = 1 - \sigma \Rightarrow H_1$.*

Let $\psi$ be any estimate of $\sigma$ that takes $Z$ to return $\psi(Z) \in \{0, 1\}$, we will prove the following two items:

① $\mathbb{P}_{\boldsymbol{I}^*}(\inf_\psi \mathbb{P}_Z(\psi(Z) \neq \sigma | \boldsymbol{I}^*) \geq c) \geq c_1$ for some universal constants $c, c_1 > 0$, namely, with probability at least $c_1$, no adaptive estimation procedure $\psi$ of $\sigma$ exists given data $Z$.

② $\mathbb{P}_{\boldsymbol{I}^*}(|\boldsymbol{I}^*| \geq t^*/10) \geq c_2$ for some universal constant $c_2 > 0$. This implies that with probability at least $c_2$, the minimax rate is at least as fast as $(nt^*)^{-1/(2-\beta)}$.

Note that if $c_1 + c_2 > 1$, there must be a multitask setting $\boldsymbol{I}^*$ from the random construction such that both ① and ② hold, and thus the above two bounds complete the proof, i.e., no adaptive learning algorithm exists under the multitask setting $\boldsymbol{I}^*$.

### 5.3. Pooling Achieves Optimal Adaptivity Sometimes

Perhaps surprising, for our designed multitask problem in this section, pooling achieves nearly optimal adaptive rates. Let us first recall the following existing pooling bound.

**Lemma 5.3** (**Hanneke & Kpotufe, 2022, Theorem 9**). *Let $Z = \bigcup_{t \in [N]} Z_t$ and $\hat{h}_{pool} = \mathrm{ERM}(Z)$. For any $\alpha \in (0, 1]$, let $t(\alpha) = \lceil \alpha \cdot N \rceil$ and $C = (32C_0^2/\alpha)^{2-\beta}C_\beta$. Then, for*

*any $\delta \in (0, 1)$, with probability at least $1 - \delta$, we have*

$$
\mathcal{E}_{\mathcal{D}}(\hat{h}_{pool}) \leq C_\rho \left( C \cdot \frac{d_{\mathcal{H}} \log\left(\frac{nN}{d_{\mathcal{H}}}\right) + \log\left(\frac{1}{\delta}\right)}{nN} \right)^{1/(2-\beta)\bar{\rho}_{t(\alpha)}} .
$$

We can think of $\alpha$ as the portion of the good datasets among all datasets. In particular, for our construction, when $N = n^{n\beta/(1-\beta)}$ and $t^* = \sqrt{n^{n\beta/(2-\beta)}N}$, we have $t(\alpha) = t^* = \Omega(\sqrt{n^{n\beta/(2-\beta)}N})$ and thus $\alpha = \Omega(N^{-1/2}n^{n\beta/2(2-\beta)})$. Plugging into Lemma 5.3, we have $C = (32C_0^2/\alpha)^{2-\beta}C_\beta$ and thus $C = O\left(N^{(2-\beta)/2}n^{-n\beta/2}\right)$. Therefore, the pooling bound implies that

$$
\mathbb{E}\left[\mathcal{E}_{\mathcal{D}}(\hat{h}_{\text{pool}})\right] \lesssim \left(\frac{C \log(nN)}{nN}\right)^{1/(2-\beta)}
$$
$$
\lesssim \left(\frac{\log(nN)}{n^{n\beta/2+1}N^{\beta/2}}\right)^{1/(2-\beta)} \lesssim \left(\frac{\log(nN)}{n\sqrt{N}}\right)^{1/(2-\beta)},
$$

where we use $\lesssim$ to hide constant factors and the last inequality follows from $N \geq n^{n\beta/(1-\beta)}$. Note that this pooling bound matches our lower bound in Theorem 5.2 up to a logarithmic factor of $\log(nN)$, implying that pooling is nearly optimal among adaptive algorithms.

## 6. Discussion and Future Research

In this paper, we investigate the hardness of adaptation for multitask learning and prove an impossibility result from an information-theoretical perspective. In brief, we show that even unlimited sample size per task does not help to realize adaptivity, that is, allowing algorithms to automatically identify the optimal aggregation of the datasets and achieve optimal rates of convergence without access to any distributional information. Our proof technique leverages an information-theoretic hypothesis testing framework. We directly bound above the KL-divergence between the likelihood functions when data are generated according to mixture distributions with opposite labels. This bound, together the celebrated Fano's method, rules out the possibility of any adaptive learner. Below, we list several technical questions that remain open in our work, as well as some interesting research problem for future work.

**Optimal Adaptivity in Multitask Learning.** Recently, Hanneke & Kpotufe (2022) has established matching upper and lower bounds for the minimax (oracle) rates of multitask learning. Such uniform rates concern the worst-case scenario over all possible distributions, but equip the learner with access to distributional information which is impractical. Consequently, designing adaptive algorithms that only have access to multisource data and can automatically leverage beneficial datasets becomes important, which relies on answering the following fundamental questions.

#### Question (Optimal Adaptivity).
- *What are the minimax optimal adaptive rates?*
- *What adaptive algorithms achieve the optimal adaptive rates?*

To formalize the definition of optimal adaptive rates, we inherit the same notations from Section 2. Let $\hat{h}_{\mathcal{A}}$ denote any adaptive multitask learner. We ask for tight upper and lower bounds (might still be in terms of $n, N$ and $\{\rho_t, t \in [N]\}$) on the following objective $\inf_{\hat{h}_{\mathcal{A}}} \sup_{\Pi \in \mathcal{M}} \mathbb{E}_{\Pi}[\mathcal{E}_{\mathcal{D}}(\hat{h}_{\mathcal{A}})]$. As for optimal adaptive algorithms, we have shown in Section 4 and Subsection 5.3 that pooling could sometimes be nearly optimal or suboptimal. It would be interesting if we could have a uniformly optimal adaptive algorithm.

**Possibility of Adaptation with A Fewer Number of Tasks.**
This work focuses on understanding whether multitask learning can be adaptive when having a large number of samples per task and provides a negative answer. A natural extension of our study is to understand whether having fewer tasks could facilitate adaptation. Indeed, it has been shown that transfer learning (with a single source task) can be adaptive (Hanneke & Kpotufe, 2019). Additionally, adaptation is possible when $N = O(\mathrm{polylog}(n))$. This can be obtained via optimizing the bound in Lemma 5.3 over the choice of $\alpha$, which gives us the following result.

**Lemma 6.1 (Hanneke & Kpotufe, 2022, Corollary 2).** *Let $C = (32C_0^2)^{2-\beta}C_\beta$. For any $\delta \in (0, 1)$, with probability at least $1 - \delta$, we have*

$$
\mathcal{E}_{\mathcal{D}}(\hat{h}_{pool}) \leq \min_{t \in [N]} C_\rho \left( C \cdot \frac{d_{\mathcal{H}} \log\left(\frac{nN}{d_{\mathcal{H}}}\right) + \log\left(\frac{1}{\delta}\right)}{(nt)^{2-\beta}(nN)^{\beta-1}} \right)^{1/(2-\beta)\bar{\rho}_t} .
$$

When $N = O(\mathrm{polylog}(n))$, Lemma 6.1 directly implies that the pooling rates are nearly optimal up to a polylogarithmic factor of $n$. In other words, pooling is almost adaptive in this situation. Moreover, all the known negative results require $N = \Omega(\exp(n))$ number of tasks.

Given the large discrepancy w.r.t. the number of tasks between existing positive and negative results, it is natural to ask what could happen under the intermediate situation. This is worth exploring as a fundamental problem of understanding the statistical limits of multitask learning. In addition, we could barely hope to have a collaboration of exponentially number of tasks in practical learning scenarios. Ideally, we hope to answer the following question: Is there a neat, quantitative threshold $N^* = \mathrm{poly}(n)$ such that adaptivity is possible if $N = O(N^*)$ and impossible whenever $N = \Omega(N^*)$? We believe that closing this gap is technically challenging, which might rely on answering the question of optimal adaptivity first.

**Adaptivity in Other Related Areas.** Adaptivity is of particular interests when facing data shortage. Multitask learning and related topics such as transfer learning, meta learning, in-context learning are such examples where target data is limited and thus knowledge transfer is pursued in order to adapt. Yet, adaptivity has not been fully understood for these models on classification, regression, and bandits problems, which could be interesting future directions.

## Acknowledgements

We thank the reviewers who provided useful suggestions on improving the quality of this paper. Steve Hanneke acknowledges support by grant no. 2024243 from the United States - Israel Binational Science Foundation (BSF).

## Impact Statement

This paper presents work whose goal is to advance the field of Statistical Machine Learning Theory. Since this work is mainly theoretical in its nature, there are no societal implications that require discloser as far as we can discern.

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

## A. Missing proofs from Section 3

*Proof of Lemma 3.1.* Under this simple setting, we only need to consider the remaining $h \in \mathcal{H} \setminus \{h^*\}$. If $P$ satisfies the Bernstein class condition, we have by definition

$$P_X(h \neq h^*) = P_X(x_1) \leq C_\beta \cdot (\mathcal{E}_P(h))^\beta = C_\beta \cdot [P(X = x_1, Y = y^*) - P(X = x_1, Y \neq y^*)]^\beta$$
$$= C_\beta \cdot [(2P(Y = y^*|X = x_1) - 1) \cdot P_X(x_1)]^\beta.$$

Simplifying the above inequality, we derive

$$P(Y = y^*|X = x_1) \geq (1/2) \cdot \left(1 + C_\beta^{-1/\beta} [P_X(x_1)]^{1/\beta - 1}\right).$$

Obviously, we know it is also a sufficient condition. $\square$

## B. Missing proofs from Section 4

*Proof of Theorem 4.1.* The proof simply follows from the multiplicative Chernoff bound. Let $\hat{n}_0(Z_P)$ and $\hat{n}_1(Z_P)$ denote the total numbers of 0 and 1 labeled examples in $Z_P$, respectively. We assume w.l.o.g. that $h^* = h_1$, i.e., $h^*(x_1) = y^* = 1$. It follows that

$$\mathbb{P}\left(\hat{h}_{Z_P} \neq h^*\right) = \mathbb{P}\left(\hat{n}_0(Z_P) \geq \hat{n}_1(Z_P)\right) = \mathbb{P}\left(\hat{n}_0(Z_P) \geq \frac{n}{2}\right).$$

Since $\hat{n}_0(Z_P)$ follows a binomial distribution $\text{Bin}(n, p)$ with $p = P(Y = 0|x_1) = 1/2 - \epsilon$, we have

$$\mathbb{P}\left(\hat{n}_0(Z_P) \geq \frac{n}{2}\right) \overset{\text{Lemma } E.2}{\leq} \exp\left(-\frac{(1 - 2p)^2 n}{4p^2 + 2p}\right) = \exp\left(-\frac{2n\epsilon^2}{(1 - \epsilon)^2}\right) \leq \exp\left(-n\epsilon^2\right) = \delta.$$

$\square$

*Proof of Theorem 4.2.* Our proof relies on an application of the Fano's inequality (Lemma E.6). Let us first restate our problem as follow. Assume that $h^*(x_1) = \sigma \in \{0, 1\}$ and consider the following two distributions

- $P = P_X \times P_{Y|X}$: $P_X(x_1) = 1$ and $P_{Y|X}(Y = \sigma|X = x_1) = 1/2 + \epsilon$.
- $Q = Q_X \times Q_{Y|X}$: $Q_X(x_1) = 1$ and $Q_{Y|X}(Y = \sigma|X = x_1) = 1/2$.

Let $\{Z_t : t \in [N]\}$ be $N$ datasets we get from sources and let $t^* \sim \text{Unif}([N])$ be the index of the dataset drawn from $P$. Let $\hat{n}_0(Z_t)$ and $\hat{n}_1(Z_t)$ denote the number of 0 and 1 labeled examples in the dataset $Z_t$ satisfying $\hat{n}_0(Z_t) + \hat{n}_1(Z_t) = n$. We further define the following statistic

$$\hat{T}_\sigma := \{\hat{n}_\sigma(Z_1), \ldots, \hat{n}_\sigma(Z_N)\}.$$

Now, it suffices to build a guarantee on any estimate $\psi$ of $\sigma$ that takes $\hat{T}_\sigma$ to return $\psi(\hat{T}_\sigma) \in \{0, 1\}$. Let $P_\sigma$ denote the probability distribution of $\hat{T}_\sigma$ for each $\sigma \in \{0, 1\}$. We can marginalize over $t^* \sim \text{Unif}([N])$ and get the likelihood function (in the remaining part of this proof, we will write $\hat{n}_{\sigma,t}$ instead of $\hat{n}_\sigma(Z_t)$ for notational simplicity)

$$P_\sigma(\hat{T}_\sigma) = \sum_{t^*=1}^N P_\sigma(\hat{T}_\sigma|t^*) P_\sigma(t^*) = \frac{1}{N} \sum_{t^*=1}^N P(\hat{n}_\sigma(Z_{t^*})) \prod_{t \neq t^*} Q(\hat{n}_\sigma(Z_t))$$
$$= \frac{1}{N} \sum_{t^*=1}^N \prod_{t=1}^N \binom{n}{\hat{n}_{\sigma,t}} \left(\frac{1}{2} + \epsilon\right)^{\hat{n}_{\sigma,t^*}} \left(\frac{1}{2} - \epsilon\right)^{n - \hat{n}_{\sigma,t^*}} \left(\frac{1}{2}\right)^{n(N-1)}. \tag{3}$$

Note that $\mathbb{D}_{\text{KL}}(P_\sigma||P_{1-\sigma}) = \mathbb{E}_{\hat{T}_\sigma \sim P_\sigma}[\log(P_\sigma(\hat{T}_\sigma)/P_{1-\sigma}(\hat{T}_\sigma))]$, our goal is to upper bound

$$\mathbb{D}_{\text{KL}}(P_\sigma||P_{1-\sigma}) \overset{(3)}{=} \mathbb{E}_{\hat{T}_\sigma \sim P_\sigma}\left[\log\left(\frac{\sum_{t=1}^N \left(\frac{1}{2} + \epsilon\right)^{\hat{n}_{\sigma,t}} \left(\frac{1}{2} - \epsilon\right)^{n - \hat{n}_{\sigma,t}}}{\sum_{t=1}^N \left(\frac{1}{2} - \epsilon\right)^{\hat{n}_{\sigma,t}} \left(\frac{1}{2} + \epsilon\right)^{n - \hat{n}_{\sigma,t}}}\right)\right]. \tag{4}$$

Since $t^*$ is uniform, the expectation above is the same as given the $t^*$, i.e., we will consider $t^*$ as fixed in the following analysis. For notational simplicity, we denote

$$V_1 := \sum_{t \neq t^*} \left(\frac{1}{2} + \epsilon\right)^{\hat{n}_{\sigma,t}} \left(\frac{1}{2} - \epsilon\right)^{n-\hat{n}_{\sigma,t}},$$

$$V_2 := \sum_{t \neq t^*} \left(\frac{1}{2} - \epsilon\right)^{\hat{n}_{\sigma,t}} \left(\frac{1}{2} + \epsilon\right)^{n-\hat{n}_{\sigma,t}},$$

$$W_1 := \left(\frac{1}{2} + \epsilon\right)^{\hat{n}_{\sigma,t^*}} \left(\frac{1}{2} - \epsilon\right)^{n-\hat{n}_{\sigma,t^*}},$$

$$W_2 := \left(\frac{1}{2} - \epsilon\right)^{\hat{n}_{\sigma,t^*}} \left(\frac{1}{2} + \epsilon\right)^{n-\hat{n}_{\sigma,t^*}}.$$

Then, we can write the RHS of (4) as

$$\mathbb{E}\left[\log\left(\frac{V_1 + W_1}{V_2 + W_2}\right)\right] = \mathbb{E}\left[\log\left(\frac{V_1 + W_1}{V_1 + W_2}\right)\right] \leq \mathbb{E}\left[\frac{W_1 - W_2}{V_1 + W_2}\right] \leq \mathbb{E}\left[\frac{W_1}{V_1}\right] = \mathbb{E}[W_1] \cdot \mathbb{E}\left[\frac{1}{V_1}\right], \tag{5}$$

where the first equation follows from that $(V_1, W_2)$ and $(V_2, W_2)$ have the same joint conditional distribution (conditional on $t^*$), the first inequality is exactly $\log x \leq x - 1$ and the last equation is because $W_1$ and $V_1$ are independent. The remaining jobs are:

**(1) Upper bound of $\mathbb{E}[W_1]$:** We bound $\mathbb{E}[W_1]$ over the randomness of $\hat{n}_{\sigma,t^*} \sim \text{Bin}(n, 1/2 + \epsilon)$.

$$\mathbb{E}[W_1] = \mathbb{E}_{\hat{n}_{\sigma,t^*} \sim \text{Bin}(n,1/2+\epsilon)}\left[\left(\frac{1}{2} + \epsilon\right)^{\hat{n}_{\sigma,t^*}} \left(\frac{1}{2} - \epsilon\right)^{n-\hat{n}_{\sigma,t^*}}\right]$$

$$= \sum_{s=0}^{n} \binom{n}{s} \left(\frac{1}{2} + \epsilon\right)^{2s} \left(\frac{1}{2} - \epsilon\right)^{2(n-s)}$$

$$= \left(\left(\frac{1}{2} + \epsilon\right)^2 + \left(\frac{1}{2} - \epsilon\right)^2\right)^n = \left(\frac{1}{2} + 2\epsilon^2\right)^n = \left(\frac{1}{2} + \frac{2}{n}\ln\left(\frac{1}{\delta}\right)\right)^n$$

$$= 2^{-n} \cdot \left(1 + \frac{4\ln(1/\delta)}{n}\right)^{\frac{n}{4\ln(1/\delta)} \cdot 4\ln(1/\delta)} \leq 2^{-n} \cdot \delta^{-4}.$$

**(2) Upper bound of $\mathbb{E}[V_1^{-1}]$:** We denote $V_1 := \sum_{t \neq t^*} g_{\sigma,t}$, where

$$g_{\sigma,t} = \left(\frac{1}{2} + \epsilon\right)^{\hat{n}_{\sigma,t}} \left(\frac{1}{2} - \epsilon\right)^{n-\hat{n}_{\sigma,t}}, \quad \forall t \neq t^*$$

are i.i.d. random variables. For every $t \neq t^*$, we further define

$$\tilde{g}_{\sigma,t} = \left(\frac{1}{2} + \epsilon\right)^{n-\hat{n}_{\sigma,t}} \left(\frac{1}{2} - \epsilon\right)^{\hat{n}_{\sigma,t}},$$

which satisfies

$$\mathbb{E}[\tilde{g}_{\sigma,t}] = \mathbb{E}_{\hat{n}_{\sigma,t} \sim \text{Bin}(n,1/2)}\left[\left(\frac{1}{2} + \epsilon\right)^{n-\hat{n}_{\sigma,t}} \left(\frac{1}{2} - \epsilon\right)^{\hat{n}_{\sigma,t}}\right]$$

$$= \sum_{s=0}^{n} \left(\frac{1}{2} + \epsilon\right)^{n-s} \left(\frac{1}{2} - \epsilon\right)^s \cdot \binom{n}{s}\left(\frac{1}{2}\right)^n = 2^{-n}.$$

We notice the following bound for every $t \neq t^*$,

$$g_{\sigma,t} \cdot \tilde{g}_{\sigma,t} = \left(\frac{1}{4} - \epsilon^2\right)^n = \left(\frac{1}{4} - \frac{1}{n}\ln\left(\frac{1}{\delta}\right)\right)^n$$

$$= 4^{-n} \cdot \left(1 - \frac{4\ln(1/\delta)}{n}\right)^{\frac{n}{4\ln(1/\delta)} \cdot 4\ln(1/\delta)}$$

$$\geq 4^{-n} \cdot \delta^4,$$

which implies the following bound

$$\mathbb{E}\left[\frac{1}{V_1}\right] = \mathbb{E}\left[\frac{1}{\sum_{t \neq t^*} g_{\sigma,t}}\right] \leq \mathbb{E}\left[\frac{1}{\sum_{t \neq t^*} \frac{4^{-n}\delta^4}{\tilde{g}_{\sigma,t}}}\right] \leq \frac{1}{N-1} \cdot \mathbb{E}\left[\frac{\sum_{t \neq t^*} \tilde{g}_{\sigma,t}}{(N-1)4^{-n}\delta^4}\right] = \frac{2^n\delta^{-4}}{N-1},$$

where the last inequality is the AM-HM inequality. Therefore, when $N \geq \delta^{-8} + 1$, we have

$$\mathbb{D}_{\mathrm{KL}}(P_\sigma || P_{1-\sigma}) \leq \mathbb{E}[W_1] \cdot \mathbb{E}\left[\frac{1}{V_1}\right] \leq 1.$$

Similarly, we can derive the same upper bound for $\mathbb{D}_{\mathrm{KL}}(P_{1-\sigma} || P_\sigma)$. Finally, according to Lemma E.6, we have

$$\inf_\psi \mathbb{P}\left(\psi(\hat{T}_\sigma) \neq \sigma\right) \geq \frac{1}{2}\left(1 - \sqrt{\frac{1}{2}\min\{\mathbb{D}_{\mathrm{KL}}(P_0 || P_1), \mathbb{D}_{\mathrm{KL}}(P_1 || P_0)\}}\right) \geq \frac{2-\sqrt{2}}{4}.$$

$\square$

*Proof of Theorem 4.3.* Let $\hat{n}_0(Z)$ and $\hat{n}_1(Z)$ denote the numbers of 0 and 1 labeled examples in $Z$, respectively. We assume w.l.o.g. that $h^* = h_1$, i.e., $h^*(x_1) = y^* = 1$. It follows that

$$\mathbb{P}\left(\hat{h}_{\mathrm{pool}} \neq h^*\right) = \mathbb{P}\left(\hat{n}_0(Z) \geq \hat{n}_1(Z)\right).$$

We can write $\hat{n}_0(Z) = \hat{n}_{0,P}(Z) + \hat{n}_{0,Q}(Z)$ where $\hat{n}_{0,P}(Z)$ denote the number of 0-labeled examples generated from distribution $P$ and $\hat{n}_{0,Q}(Z)$ is defined similarly. First, note that $\hat{n}_{0,P}(Z) \sim \mathrm{Bin}(n,p)$ with $p = P(Y = 0|x_1) = 1/2 - \epsilon$, we know that at the most extreme case $\hat{n}_{1,P}(Z) - \hat{n}_{0,P}(Z) \leq n$. Moreover, recall that $\hat{n}_{0,Q}(Z) \sim \mathrm{Bin}(Nn, 1/2)$ (it is indeed $\mathrm{Bin}((N-1)n, 1/2)$ and assuming $\mathrm{Bin}(Nn, 1/2)$ makes no difference asymptotically). We can write $\hat{n}_{0,Q}(Z) = \sum_{j \in [nN]} \mathbb{1}\{Y_j = 0\}$ where $Y_j \sim \mathrm{Ber}(1/2)$ for any $j \in [nN]$. It holds that $\mathbb{E}[\mathbb{1}\{Y_j = 0\}] = 1/2$, $\mathrm{Var}(\mathbb{1}\{Y_j = 0\}) = 1/4$ and $\mathbb{E}[|\mathbb{1}\{Y_j = 0\}|^3] = 1/2$. By Berry-Esseen inequality (Lemma E.8), we have

$$\left|\mathbb{P}\left(\frac{2\hat{n}_{0,Q}(Z)}{\sqrt{nN}} - \sqrt{nN} \leq x\right) - \Phi(x)\right| \leq \frac{12}{\sqrt{nN}}.$$

Hence, choosing $x = 2\sqrt{n/N}$ yields that

$$\mathbb{P}\left(\hat{n}_{0,Q}(Z) - \frac{nN}{2} \leq n\right) - \Phi\left(2\sqrt{\frac{n}{N}}\right) \leq \left|\mathbb{P}\left(\hat{n}_{0,Q}(Z) - \frac{nN}{2} \leq n\right) - \Phi\left(2\sqrt{\frac{n}{N}}\right)\right| \leq \frac{12}{\sqrt{nN}},$$

which implies immediately that

$$\mathbb{P}\left(\hat{n}_{0,Q}(Z) \geq \frac{nN}{2} + n\right) \geq 1 - \Phi\left(2\sqrt{\frac{n}{N}}\right) - \frac{12}{\sqrt{nN}}.$$

When $n$ is large enough and $N \geq 4n$, we have

$$\mathbb{P}\left(\hat{n}_{0,Q}(Z) \geq \frac{nN}{2} + n\right) \geq 1 - \Phi(1) - \frac{12}{\sqrt{nN}} \geq 0.1.$$

Putting together, we have with probability at least $0.1$, pooling predicts incorrectly since

$$\hat{n}_0(Z) - \hat{n}_1(Z) = \hat{n}_{0,P}(Z) - \hat{n}_{1,P}(Z) + \hat{n}_{0,Q}(Z) - \hat{n}_{1,Q}(Z) \geq -n + n = 0.$$

Therefore, we get $\mathbb{E}[\mathcal{E}_\mathcal{D}(\hat{h}_{\mathrm{pool}})] \geq 0.1\epsilon = \Omega(1)$. $\square$

*Proof of Theorem 4.4.* By Lemma E.11 and union bound, we know that $h^* \in \bigcap_{t=1}^{N} \mathcal{H}_t$ with probability at least $1 - \sum_{t \in [N]} \delta_t \geq 1 - \delta$. Since our construction has only $\mathcal{H} = \{h^*, h\}$, it suffices to show that $\bigcap_{t=1}^{N} \mathcal{H}_t = \{h^*\}$, i.e., there exists $t \in [N]$ such that $\mathcal{H}_t = \{h^*\}$ with high probability. Consider the $\mathcal{H}_{t^*}$ associated with the good dataset $Z_{t^*} \sim P$. By Lemma E.10, if $h \in \mathcal{H}_{t^*}$, then with probability at least $1 - \delta_{t^*} \geq 1 - \delta/6$, we have

$$\mathcal{E}_P(h) \leq 32 C_0^2 \left( C_\beta \cdot \epsilon(n, \delta_{t^*}) \right)^{1/(2-\beta)} \overset{\beta=0}{=} O\left( \sqrt{\frac{d_{\mathcal{H}}}{n} \log\left(\frac{n}{d_{\mathcal{H}}}\right) + \frac{1}{n}\log\left(\frac{N}{\delta}\right)} \right).$$

However, recall that $\mathcal{E}_P(h) = 2\epsilon = \Omega(1)$ in our construction. In other words, we can guarantee that $h \notin \mathcal{H}_{t^*}$ and thus $\mathcal{H}_{t^*} = \{h^*\}$ with probability at least $1 - 2\delta$, if $\log(N/\delta) = o(n)$ (for example, assuming $N = \Theta(n)$ and $\delta = 1/n$ would be sufficient both here and in Theorem 4.3). It is clear that when $\mathcal{H}_{t^*} = \{h^*\}$ and thus $\bigcap_{t=1}^{N} \mathcal{H}_t = \{h^*\}$, we have $\mathcal{E}_{\mathcal{D}}(\hat{h}_{\mathrm{IB}}) = \mathcal{E}_{\mathcal{D}}(h^*) = 0$. Altogether, choosing $\delta = 1/n$ yields

$$\mathbb{E}\left[\mathcal{E}_{\mathcal{D}}(\hat{h}_{\mathrm{IB}})\right] \leq 2\delta\epsilon = O\left(\frac{1}{n}\right).$$

Indeed, given the following sequence of transfer exponents $\{1, \infty, \ldots, \infty\}$, we can show a $\sqrt{\log(N)/n}$ upper bound on the learning rates of Algorithm 1. Let us consider any distributions $P_1, P_2, \ldots, P_N$ such that $\rho_{t^*} = 1$ and $\rho_{\neq t^*} = \infty$ for some $t^* \in [N]$. Since $\hat{h}_{\mathrm{IB}} \in \mathcal{H}$, we have by definition

$$\mathcal{E}_{P_{t^*}}(\hat{h}_{\mathrm{IB}}) \geq \left( C_\rho^{-1} \mathcal{E}_{\mathcal{D}}(\hat{h}_{\mathrm{IB}}) \right)^{\rho_{t^*}} = C_\rho^{-1} \mathcal{E}_{\mathcal{D}}(\hat{h}_{\mathrm{IB}}) \Rightarrow \mathcal{E}_{\mathcal{D}}(\hat{h}_{\mathrm{IB}}) \leq C_\rho \mathcal{E}_{P_{t^*}}(\hat{h}_{\mathrm{IB}}). \tag{6}$$

Next, by Lemma E.11 and union bound, we know that $h^* \in \bigcap_{t=1}^{N} \mathcal{H}_t$, that is, for every $t \in [N]$,

$$\hat{\mathcal{E}}_{Z_t}(h^*, \hat{h}_{Z_t}) \leq C_0 \sqrt{\hat{P}_{Z_t}(h^* \neq \hat{h}_{Z_t}) \cdot \epsilon(n, \delta_t)} + C_0 \cdot \epsilon(n, \delta_t),$$

with probability at least $1 - \sum_{t \in [N]} \delta_t \geq 1 - \delta$. In other words, Algorithm 1 can always at least output $h^*$. By Lemma E.10, we know that $\hat{h}_{\mathrm{IB}}$ satisfies

$$\mathcal{E}_{P_{t^*}}(\hat{h}_{\mathrm{IB}}) \leq 32 C_0^2 \left( C_\beta \cdot \epsilon(n, \delta_{t^*}) \right)^{1/(2-\beta)} \overset{\beta=0}{\leq} \tilde{C} \sqrt{\frac{d_{\mathcal{H}}}{n} \log\left(\frac{n}{d_{\mathcal{H}}}\right) + \frac{1}{n}\log\left(\frac{N}{\delta}\right)}$$

for some numerical constant $\tilde{C} > 0$. Together with (6), we have with probability at least $1 - \delta$,

$$\mathcal{E}_{\mathcal{D}}(\hat{h}_{\mathrm{IB}}) \leq \tilde{C} C_\rho \sqrt{\frac{d_{\mathcal{H}}}{n} \log\left(\frac{n}{d_{\mathcal{H}}}\right) + \frac{1}{n}\log\left(\frac{N}{\delta}\right)}.$$

Recall that for any non-negative random variable $Z$, $\mathbb{E}[Z] = \int_0^\infty \mathbb{P}(Z > t)dt$. Hence, we have

$$\mathbb{E}\left[\mathcal{E}_{\mathcal{D}}(\hat{h}_{\mathrm{IB}})\right] = \int_0^\infty \mathbb{P}\left(\mathcal{E}_{\mathcal{D}}(\hat{h}_{\mathrm{IB}}) > t\right) dt \leq \int_0^\infty \min\left\{ \frac{N\left(\frac{n}{d_{\mathcal{H}}}\right)^{d_{\mathcal{H}}}}{\exp\left(\frac{nt^2}{\tilde{C}^2 C_\rho^2}\right)}, 1 \right\} dt$$

$$\leq \int_0^{\tilde{C}C_\rho \sqrt{\frac{d_{\mathcal{H}}}{n}\log\left(\frac{n}{d_{\mathcal{H}}}\right) + \frac{\log N}{n}}} dt + \int_{\tilde{C}C_\rho\sqrt{\frac{d_{\mathcal{H}}}{n}\log\left(\frac{n}{d_{\mathcal{H}}}\right) + \frac{\log N}{n}}}^{\infty} \frac{N\left(\frac{n}{d_{\mathcal{H}}}\right)^{d_{\mathcal{H}}}}{\exp\left(\frac{nt^2}{\tilde{C}^2 C_\rho^2}\right)} dt$$

$$\leq \tilde{C}C_\rho \sqrt{\frac{d_{\mathcal{H}}}{n}\log\left(\frac{n}{d_{\mathcal{H}}}\right) + \frac{\log N}{n}} + o\left(\sqrt{\frac{d_{\mathcal{H}}}{n}\log\left(\frac{n}{d_{\mathcal{H}}}\right) + \frac{\log N}{n}}\right)$$

$$= O\left(\sqrt{\frac{\log N}{n}}\right),$$

where the last inequality uses the fact that $\int_a^\infty e^{-ct^2} dt \leq \frac{e^{-ca^2}}{2ca}$ for any $a > 0$. $\qquad\square$

# C. Missing proofs from Section 5

*Proof of Theorem 5.1.* Let $\hat{n}_0(Z^*)$ and $\hat{n}_1(Z^*)$ denote the total number of 0 and 1 labeled examples in $Z^*$ with marginals at $x_1$, i.e.,

$$\hat{n}_\sigma(Z^*) := \sum_{t \in [t^*]} \sum_{i \in Z_{(t)}} \mathbb{1}\left\{ x_{t,i} = x_1 \wedge y_{t,i} = \sigma \right\}, \ \forall \sigma \in \{0, 1\},$$

and $\hat{n}_{x_1}(Z^*) = \hat{n}_0(Z^*) + \hat{n}_1(Z^*)$ denote the total number of examples in $Z^*$ with marginals at $x_1$.

Assume w.l.o.g. $h^* = h_1$, that is, $h^*(x_1) = y^* = 1$. Then ERM makes a mistake on $x_1$ if and only if we have less number of samples with label 1 than label 0, that is

$$\mathbb{P}\left( \hat{h}_{Z^*} \neq h^* \right) = \mathbb{P}\left( \hat{n}_0(Z^*) \geq \hat{n}_1(Z^*) \right) = \mathbb{P}\left( \hat{n}_0(Z^*) \geq \frac{\hat{n}_{x_1}(Z^*)}{2} \right). \tag{7}$$

Since for every $t \in [t^*]$, $P_{(t)} = P$, the dataset $Z^*$ can be considered to be drawn i.i.d. from $P$. We can apply Chernoff bound (Lemma E.2) with $p = P(Y = 0 | X = x_1)$, $m = \hat{n}_{x_1}(Z^*)$ and $\delta = 1/2p - 1$ (note that $p < 1/2$ so that $\delta > 0$ is satisfied), and get

$$\mathbb{P}\left( \hat{n}_0(Z^*) \geq \frac{\hat{n}_{x_1}(Z^*)}{2} \right) \leq \exp\left( -\frac{(1 - 2p)^2 \cdot \hat{n}_{x_1}(Z^*)}{4p + 2} \right). \tag{8}$$

Based on Lemma 3.1 and our construction, we have

$$p = 1 - P(Y = 1 | X = x_1) \leq 1/2 - (1/2)C_\beta^{-1/\beta} \left[ P_X(x_1) \right]^{1/\beta - 1}. \tag{9}$$

Note that the RHS of (8) is monotonically increasing as a function of $p$, we have from (9) that

$$\mathbb{P}\left( \hat{n}_0(Z^*) \geq \frac{\hat{n}_{x_1}(Z^*)}{2} \right) \leq \exp\left\{ -\frac{C_\beta^{-2/\beta} \left[ P_X(x_1) \right]^{2/\beta - 2} \cdot \hat{n}_{x_1}(Z^*)}{4 - 2C_\beta^{-1/\beta} \left[ P_X(x_1) \right]^{1/\beta - 1}} \right\}. \tag{10}$$

To quantify $\hat{n}_{x_1}(Z^*)$, we will still use Lemma E.2 with $m = nt^*$, $p = P_X(x_1)$ and $\delta = 1/2$,

$$\mathbb{P}\left( \hat{n}_{x_1}(Z^*) \leq \frac{nt^* P_X(x_1)}{2} \right) \leq \exp\left\{ -\frac{nt^* P_X(x_1)}{8} \right\}. \tag{11}$$

Using the fact that $\mathbb{P}(A) \leq \mathbb{P}(A|B) + \mathbb{P}(\neg B)$, we have

$$\mathbb{P}\left( \hat{n}_0(Z^*) \geq \frac{\hat{n}_{x_1}(Z^*)}{2} \right)$$

$$\leq \ \mathbb{P}\left( \hat{n}_0(Z^*) \geq \frac{\hat{n}_{x_1}(Z^*)}{2} \ \middle| \ \hat{n}_{x_1}(Z^*) \geq \frac{nt^* P_X(x_1)}{2} \right) + \mathbb{P}\left( \hat{n}_{x_1}(Z^*) \leq \frac{nt^* P_X(x_1)}{2} \right)$$

$$\overset{(10),(11)}{\leq} \ \exp\left\{ -nt^* \left( \frac{C_\beta^{-2/\beta} \left[ P_X(x_1) \right]^{2/\beta - 1}}{8 - 4C_\beta^{-1/\beta} \left[ P_X(x_1) \right]^{1/\beta - 1}} \right) \right\} + \exp\left\{ -\frac{nt^* P_X(x_1)}{8} \right\}$$

$$\leq \ \exp\left\{ -\frac{nt^* C_\beta^{-2/\beta} \left[ P_X(x_1) \right]^{2/\beta - 1}}{8} \right\} + \exp\left\{ -\frac{nt^* P_X(x_1)}{8} \right\}. \tag{12}$$

Note that $P_X(x_1) = C_\beta \epsilon^\beta = C_\beta(n\sqrt{N})^{-\beta/(2-\beta)}$, and thus $C_\beta^{-2/\beta}[P_X(x_1)]^{2/\beta - 1} = C_\beta^{-1}(n\sqrt{N})^{-1}$. Recall that $t^* = \sqrt{N} \cdot n^{n\beta/(2-\beta)}$, it follows that

$$\text{RHS of (12)} \ = \ \exp\left\{ -\frac{t^*}{8C_\beta\sqrt{N}} \right\} + \exp\left\{ -\frac{nt^* C_\beta(n\sqrt{N})^{-\beta/(2-\beta)}}{8} \right\}$$

$$\overset{\beta \leq 1}{\leq} \ \exp\left\{ -\frac{t^*}{8C_\beta\sqrt{N}} \right\} + \exp\left\{ -\frac{C_\beta t^*}{8\sqrt{N}} \right\} \leq 2\exp\left\{ -\frac{n^{n\beta/2(2-\beta)}}{8C_\beta} \right\},$$

that is, with probability at least $1 - 2\exp\{-(8C_\beta)^{-1}n^{n\beta/2(2-\beta)}\}$, $\hat{h}_{Z^*} = h^*$ and thus $\mathcal{E}_\mathcal{D}(\hat{h}_{Z^*}) = 0$. Note that this implies that pooling over all the fair datasets achieves the minimax rates:

$$
\mathbb{E}\left[\mathcal{E}_\mathcal{D}(\hat{h}_{Z^*})\right] \leq 0 \cdot \left(1 - 2\exp\left\{-\frac{n^{n\beta/2(2-\beta)}}{8C_\beta}\right\}\right) + \epsilon \cdot 2\exp\left\{-\frac{n^{n\beta/2(2-\beta)}}{8C_\beta}\right\}
$$

$$
= \left(n\sqrt{N}\right)^{-1/(2-\beta)} \cdot 2\exp\left\{-\frac{n^{n\beta/2(2-\beta)}}{8C_\beta}\right\}.
$$

$\square$

*Proof of Theorem 5.2.* For $t \in [N]$ and $\sigma \in \{0,1\}$, we write $\hat{n}_t$ for short of $\hat{n}_{x_1}(Z_t)$ and $\hat{n}_{\sigma,t}$ for short of $\hat{n}_\sigma(Z_t)$. We further make the following simplified notations:

$$
\begin{aligned}
p_{x_1} &= P(X = x_1), \ p_{\sigma|x_1} = P(Y = \sigma | X = x_1), \ p_{\sigma,x_1} = P(Y = \sigma, X = x_1) = p_{\sigma|x_1} \cdot p_{x_1}; \\
q_{x_1} &= Q(X = x_1), \ q_{\sigma|x_1} = Q(Y = \sigma | X = x_1), \ q_{\sigma,x_1} = Q(Y = \sigma, X = x_1) = q_{\sigma|x_1} \cdot q_{x_1}; \\
p_{1,x_0} &= P(Y = 1, X = x_0), \ q_{1,x_0} = Q(Y = 1, X = x_0).
\end{aligned}
$$

To prove ①, we have from the law of total probability that

$$
\mathbb{P}\left(\psi(Z) \neq \sigma\right) = \mathbb{E}_{\boldsymbol{I}^*}\left[\mathbb{P}_Z\left(\psi(Z) \neq \sigma | \boldsymbol{I}^*\right)\right] \leq c + \mathbb{P}_{\boldsymbol{I}^*}\left(\mathbb{P}_Z\left(\psi(Z) \neq \sigma | \boldsymbol{I}^*\right) \geq c\right),
$$

which implies that

$$
\mathbb{P}_{\boldsymbol{I}^*}\left(\mathbb{P}_Z\left(\psi(Z) \neq \sigma | \boldsymbol{I}^*\right) \geq c\right) \geq \mathbb{P}\left(\psi(Z) \neq \sigma\right) - c.
$$

Hence, it suffices to lower bound $\mathbb{P}(\psi(Z) \neq \sigma)$, where the probability includes all the randomness defined in the problem. We will apply the Fano's inequality (Lemma E.6) to build such a lower bound for any estimate $\psi$, i.e., to lower bound $\inf_\psi \mathbb{P}(\psi(Z) \neq \sigma)$. We begin by calculating the likelihood of getting data $Z$ from distribution $P_\sigma$:

$$
P_\sigma(Z) = \prod_{t=1}^N P_\sigma(Z_t) = \prod_{t=1}^N \left(P_\sigma(Z_t | t \in \boldsymbol{I}^*)P_\sigma(t \in \boldsymbol{I}^*) + P_\sigma(Z_t | t \notin \boldsymbol{I}^*)P_\sigma(t \notin \boldsymbol{I}^*)\right)
$$

$$
= \prod_{t=1}^N \left(\alpha_F \cdot (p_{\sigma,x_1})^{\hat{n}_{\sigma,t}}(p_{1-\sigma,x_1})^{\hat{n}_t - \hat{n}_{\sigma,t}}(p_{1,x_0})^{n-\hat{n}_t} + \alpha_N \cdot (q_{\sigma,x_1})^{\hat{n}_{\sigma,t}}(q_{1-\sigma,x_1})^{\hat{n}_t - \hat{n}_{\sigma,t}}(q_{1,x_0})^{n-\hat{n}_t}\right),
$$

and similarly from $P_{1-\sigma}$:

$$
P_{1-\sigma}(Z) = \prod_{t=1}^N \left(\alpha_F \cdot (p_{\sigma,x_1})^{\hat{n}_t - \hat{n}_{\sigma,t}}(p_{1-\sigma,x_1})^{\hat{n}_{\sigma,t}}(p_{1,x_0})^{n-\hat{n}_t} + \alpha_N \cdot (q_{\sigma,x_1})^{\hat{n}_t - \hat{n}_{\sigma,t}}(q_{1-\sigma,x_1})^{\hat{n}_{\sigma,t}}(q_{1,x_0})^{n-\hat{n}_t}\right).
$$

Then we can write the KL-divergence between their likelihoods as

$$
\mathbb{D}_{\mathrm{KL}}(P_\sigma || P_{1-\sigma}) = \mathbb{E}_{Z \sim P_\sigma}\left[\log\left(\frac{P_\sigma(Z)}{P_{1-\sigma}(Z)}\right)\right] = \mathbb{E}\left[\log\left(\prod_{t=1}^N \frac{P_\sigma(Z_t)}{P_{1-\sigma}(Z_t)}\right)\right]
$$

$$
= \sum_{t=1}^N \mathbb{E}\left[\log\left(\frac{\alpha_F \cdot (p_{\sigma,x_1})^{\hat{n}_{\sigma,t}}(p_{1-\sigma,x_1})^{\hat{n}_t - \hat{n}_{\sigma,t}}(p_{x_0})^{n-\hat{n}_t} + \alpha_N \cdot (q_{\sigma,x_1})^{\hat{n}_{\sigma,t}}(q_{1-\sigma,x_1})^{\hat{n}_t - \hat{n}_{\sigma,t}}(q_{x_0})^{n-\hat{n}_t}}{\alpha_F \cdot (p_{\sigma,x_1})^{\hat{n}_t - \hat{n}_{\sigma,t}}(p_{1-\sigma,x_1})^{\hat{n}_{\sigma,t}}(p_{x_0})^{n-\hat{n}_t} + \alpha_N \cdot (q_{\sigma,x_1})^{\hat{n}_t - \hat{n}_{\sigma,t}}(q_{1-\sigma,x_1})^{\hat{n}_{\sigma,t}}(q_{x_0})^{n-\hat{n}_t}}\right)\right]
$$

$$
=: \sum_{t=1}^N \mathbb{E}\left[\log\left(\frac{V_{F,\sigma,t} + V_{N,\sigma,t}}{V_{F,1-\sigma,t} + V_{N,1-\sigma,t}}\right)\right] = N \cdot \mathbb{E}\left[\log\left(\frac{V_{F,\sigma,t} + V_{N,\sigma,t}}{V_{F,1-\sigma,t} + V_{N,1-\sigma,t}}\right)\right]. \tag{13}
$$

We first make the following decomposition

$$
\log\left(\frac{V_{F,\sigma,t} + V_{N,\sigma,t}}{V_{F,1-\sigma,t} + V_{N,1-\sigma,t}}\right) = \log\left(\frac{1 + V_{F,\sigma,t}/V_{N,\sigma,t}}{1 + V_{F,1-\sigma,t}/V_{N,1-\sigma,t}}\right) + \log\left(\frac{V_{N,\sigma,t}}{V_{N,1-\sigma,t}}\right). \tag{14}
$$

Note that

$$\log\left(\frac{V_{N,\sigma,t}}{V_{N,1-\sigma,t}}\right) = \log\left(\frac{\alpha_N \cdot (q_{\sigma,x_1})^{\hat{n}_{\sigma,t}}(q_{1-\sigma,x_1})^{\hat{n}_t - \hat{n}_{\sigma,t}}(q_{1,x_0})^{n-\hat{n}_t}}{\alpha_N \cdot (q_{\sigma,x_1})^{\hat{n}_t - \hat{n}_{\sigma,t}}(q_{1-\sigma,x_1})^{\hat{n}_{\sigma,t}}(q_{1,x_0})^{n-\hat{n}_t}}\right) = (2\hat{n}_{\sigma,t} - \hat{n}_t)\log\left(\frac{q_{\sigma,x_1}}{q_{1-\sigma,x_1}}\right).$$

Plugging into (14) and then into (13), we have

$$\mathbb{D}_{\mathrm{KL}}(P_\sigma||P_{1-\sigma}) = N \cdot \mathbb{E}\left[\log\left(\frac{1 + V_{F,\sigma,t}/V_{N,\sigma,t}}{1 + V_{F,1-\sigma,t}/V_{N,1-\sigma,t}}\right) + (2\hat{n}_{\sigma,t} - \hat{n}_t) \cdot \log\left(\frac{q_{\sigma,x_1}}{q_{1-\sigma,x_1}}\right)\right]. \tag{15}$$

If $Z_t \sim P^n$, then $\hat{n}_{\sigma,t} \sim \mathrm{Bin}(\hat{n}_t, p_{\sigma|x_1})$ and $\hat{n}_t \sim \mathrm{Bin}(n, p_{x_1})$; if $Z_t \sim Q^n$, then $\hat{n}_{\sigma,t} \sim \mathrm{Bin}(\hat{n}_t, q_{\sigma|x_1})$ and $\hat{n}_t \sim \mathrm{Bin}(n, q_{x_1})$. We have from the law of total expectation that

$$\begin{aligned}
\mathbb{E}\left[2\hat{n}_{\sigma,t} - \hat{n}_t\right] &= \alpha_F \cdot \mathbb{E}\left[2\hat{n}_{\sigma,t} - \hat{n}_t | Z_t \sim P^n\right] + \alpha_N \cdot \mathbb{E}\left[2\hat{n}_{\sigma,t} - \hat{n}_t | Z_t \sim Q^n\right] \\
&= \alpha_F \cdot \mathbb{E}_{\hat{n}_t}\left[\mathbb{E}_{P|\hat{n}_t}\left[2\hat{n}_{\sigma,t} - \hat{n}_t | \hat{n}_t\right]\right] + \alpha_N \cdot \mathbb{E}_{\hat{n}_t}\left[\mathbb{E}_{Q|\hat{n}_t}\left[2\hat{n}_{\sigma,t} - \hat{n}_t | \hat{n}_t\right]\right] \\
&= \alpha_F \cdot \mathbb{E}_{P,\hat{n}_t}\left[(2p_{\sigma|x_1} - 1)\hat{n}_t\right] + \alpha_N \cdot \mathbb{E}_{Q,\hat{n}_t}\left[(2q_{\sigma|x_1} - 1)\hat{n}_t\right] \\
&= \alpha_F \cdot (2p_{\sigma|x_1} - 1)np_{x_1} + \alpha_N \cdot (2q_{\sigma|x_1} - 1)nq_{x_1} \\
&= \alpha_F \cdot n\epsilon + \alpha_N \cdot n\epsilon_0.
\end{aligned}$$

The second summand in (15) can be bounded by

$$\begin{aligned}
N \cdot \mathbb{E}\left[(2\hat{n}_{\sigma,t} - \hat{n}_t) \cdot \log\left(\frac{q_{\sigma,x_1}}{q_{1-\sigma,x_1}}\right)\right] &= N \cdot \log\left(\frac{q_{\sigma,x_1}}{q_{1-\sigma,x_1}}\right) \cdot (\alpha_F \cdot n\epsilon + \alpha_N \cdot n\epsilon_0) \\
&= N \cdot \log\left(\frac{\frac{1}{2}C_\beta \epsilon_0^\beta + \frac{1}{2}\epsilon_0}{\frac{1}{2}C_\beta \epsilon_0^\beta - \frac{1}{2}\epsilon_0}\right) \cdot (\alpha_F \cdot n\epsilon + \alpha_N \cdot n\epsilon_0) \\
&\leq N \cdot \frac{2\epsilon_0}{C_\beta \epsilon_0^\beta - \epsilon_0} \cdot (\alpha_F \cdot n\epsilon + \alpha_N \cdot n\epsilon_0) \\
&\leq N\epsilon_0^{1-\beta} \cdot (\alpha_F \cdot n\epsilon + \alpha_N \cdot n\epsilon_0) \\
&= \alpha_F \cdot Nn\epsilon\epsilon_0^{1-\beta} + \alpha_N \cdot Nn\epsilon_0^{2-\beta}, \tag{16}
\end{aligned}$$

where the first inequality uses the fact that $\log x \leq x - 1$. This implies the following two bounds:

$$\mathbb{E}_P\left[\log\left(\frac{V_{N,\sigma,t}}{V_{N,1-\sigma,t}}\right)\right] \leq n\epsilon\epsilon_0^{1-\beta}, \quad \mathbb{E}_Q\left[\log\left(\frac{V_{N,\sigma,t}}{V_{N,1-\sigma,t}}\right)\right] \leq n\epsilon_0^{2-\beta}. \tag{17}$$

For the first summand in (15), based on the law of total expectation, we have

$$\begin{aligned}
&N \cdot \mathbb{E}\left[\log\left(\frac{1 + V_{F,\sigma,t}/V_{N,\sigma,t}}{1 + V_{F,1-\sigma,t}/V_{N,1-\sigma,t}}\right)\right] \\
&= N \cdot \alpha_F \cdot \mathbb{E}_P\left[\log\left(\frac{1 + V_{F,\sigma,t}/V_{N,\sigma,t}}{1 + V_{F,1-\sigma,t}/V_{N,1-\sigma,t}}\right)\right] + N \cdot \alpha_N \cdot \mathbb{E}_Q\left[\log\left(\frac{1 + V_{F,\sigma,t}/V_{N,\sigma,t}}{1 + V_{F,1-\sigma,t}/V_{N,1-\sigma,t}}\right)\right]. \tag{18}
\end{aligned}$$

We first bound the second summand in (18).

$$\begin{aligned}
\mathbb{E}_Q\left[\log\left(\frac{1 + V_{F,\sigma,t}/V_{N,\sigma,t}}{1 + V_{F,1-\sigma,t}/V_{N,1-\sigma,t}}\right)\right] &\overset{\log x \leq x - 1}{\leq} \mathbb{E}_Q\left[\frac{V_{F,\sigma,t}/V_{N,\sigma,t} - V_{F,1-\sigma,t}/V_{N,1-\sigma,t}}{1 + V_{F,1-\sigma,t}/V_{N,1-\sigma,t}}\right] \\
&\leq \max\left\{0, \mathbb{E}_Q\left[\frac{V_{F,\sigma,t}}{V_{N,\sigma,t}} - \frac{V_{F,1-\sigma,t}}{V_{N,1-\sigma,t}}\right]\right\}.
\end{aligned}$$

Note that

$$\mathbb{E}_Q\left[\frac{V_{F,\sigma,t}}{V_{N,\sigma,t}}\right] = \mathbb{E}_{\hat{n}_t}\left[\mathbb{E}_{Q|\hat{n}_t}\left[\frac{V_{F,\sigma,t}}{V_{N,\sigma,t}}\bigg|\hat{n}_t\right]\right]$$

$$=\frac{\alpha_F}{\alpha_N}\cdot\mathbb{E}_{Q,\hat{n}_t}\left[\sum_{s=0}^{\hat{n}_t}\left(\frac{p_{\sigma,x_1}}{q_{\sigma,x_1}}\right)^s\left(\frac{p_{1-\sigma,x_1}}{q_{1-\sigma,x_1}}\right)^{\hat{n}_t-s}\left(\frac{p_{1,x_0}}{q_{1,x_0}}\right)^{n-\hat{n}_t}\binom{\hat{n}_t}{s}(q_{\sigma|x_1})^s(q_{1-\sigma|x_1})^{\hat{n}_t-s}\right]$$

$$=\frac{\alpha_F}{\alpha_N}\cdot\mathbb{E}_{P,\hat{n}_t}\left[\left(\frac{p_{1,x_0}}{q_{1,x_0}}\right)^{n-\hat{n}_t}\sum_{s=0}^{\hat{n}_t}\binom{\hat{n}_t}{s}\left(\frac{p_{\sigma,x_1}q_{\sigma|x_1}}{q_{\sigma,x_1}}\right)^s\left(\frac{p_{1-\sigma,x_1}q_{1-\sigma|x_1}}{q_{1-\sigma,x_1}}\right)^{\hat{n}_t-s}\right]$$

$$=\frac{\alpha_F}{\alpha_N}\cdot\mathbb{E}_{P,\hat{n}_t}\left[\left(\frac{p_{1,x_0}}{q_{1,x_0}}\right)^{n-\hat{n}_t}\left(\frac{p_{\sigma,x_1}q_{\sigma|x_1}}{q_{\sigma,x_1}}+\frac{p_{1-\sigma,x_1}q_{1-\sigma|x_1}}{q_{1-\sigma,x_1}}\right)^{\hat{n}_t}\right]$$

$$=\frac{\alpha_F}{\alpha_N}\cdot\sum_{s=0}^{n}\left(\frac{p_{1,x_0}}{q_{1,x_0}}\right)^{n-s}\left(\frac{p_{\sigma,x_1}q_{\sigma|x_1}}{q_{\sigma,x_1}}+\frac{p_{1-\sigma,x_1}q_{1-\sigma|x_1}}{q_{1-\sigma,x_1}}\right)^s\binom{n}{s}(q_{x_1})^s(q_{x_0})^{n-s}$$

$$=\frac{\alpha_F}{\alpha_N}\cdot(p_{\sigma,x_1}+p_{1-\sigma,x_1}+p_{x_0})^n=\frac{\alpha_F}{\alpha_N},$$

and similarly

$$\mathbb{E}_Q\left[\frac{V_{F,1-\sigma,t}}{V_{N,1-\sigma,t}}\right] =\frac{\alpha_F}{\alpha_N}\cdot\left(\frac{p_{\sigma,x_1}q_{1-\sigma,x_1}}{q_{\sigma,x_1}}+\frac{p_{1-\sigma,x_1}q_{\sigma,x_1}}{q_{1-\sigma,x_1}}+p_{1,x_0}\right)^n$$

$$=\frac{\alpha_F}{\alpha_N}\cdot\left(1-p_{\sigma,x_1}\left(1-\frac{q_{1-\sigma,x_1}}{q_{\sigma,x_1}}\right)-p_{1-\sigma,x_1}\left(1-\frac{q_{\sigma,x_1}}{q_{1-\sigma,x_1}}\right)\right)^n$$

$$=\frac{\alpha_F}{\alpha_N}\cdot\left(1-\epsilon_0\left(\frac{C_\beta\epsilon^\beta+\epsilon}{C_\beta\epsilon_0^\beta+\epsilon_0}-\frac{C_\beta\epsilon^\beta-\epsilon}{C_\beta\epsilon_0^\beta-\epsilon_0}\right)\right)^n$$

$$\geq\frac{\alpha_F}{\alpha_N}\cdot\left(1-n\epsilon_0\left(\frac{C_\beta\epsilon^\beta+\epsilon}{C_\beta\epsilon_0^\beta+\epsilon_0}-\frac{C_\beta\epsilon^\beta-\epsilon}{C_\beta\epsilon_0^\beta-\epsilon_0}\right)\right),$$

where we use the Bernoulli's inequality for the last step. Note that

$$\frac{C_\beta\epsilon^\beta+\epsilon}{C_\beta\epsilon_0^\beta+\epsilon_0}-\frac{C_\beta\epsilon^\beta-\epsilon}{C_\beta\epsilon_0^\beta-\epsilon_0}=\frac{2C_\beta(\epsilon\epsilon_0^\beta-\epsilon_0\epsilon^\beta)}{C_\beta^2\epsilon_0^{2\beta}-\epsilon_0^2}=O\left(\frac{\epsilon}{\epsilon_0^\beta}\right).$$

Hence, it follows that

$$N\cdot\alpha_N\cdot\mathbb{E}_Q\left[\log\left(\frac{1+V_{F,\sigma,t}/V_{N,\sigma,t}}{1+V_{F,1-\sigma,t}/V_{N,1-\sigma,t}}\right)\right]\leq N\cdot\alpha_N\cdot\mathbb{E}_Q\left[\frac{V_{F,\sigma,t}}{V_{N,\sigma,t}}-\frac{V_{F,1-\sigma,t}}{V_{N,1-\sigma,t}}\right]$$

$$\leq N\cdot\alpha_N\cdot\frac{\alpha_F}{\alpha_N}\cdot n\epsilon_0\left(\frac{C_\beta\epsilon^\beta+\epsilon}{C_\beta\epsilon_0^\beta+\epsilon_0}-\frac{C_\beta\epsilon^\beta-\epsilon}{C_\beta\epsilon_0^\beta-\epsilon_0}\right)=O\left(\alpha_F\cdot Nn\epsilon\epsilon_0^{1-\beta}\right).\tag{19}$$

From (17) and (19), we can conclude that

$$N\cdot\alpha_N\cdot\mathbb{E}_Q\left[\log\left(\frac{V_{F,\sigma,t}+V_{N,\sigma,t}}{V_{F,1-\sigma,t}+V_{N,1-\sigma,t}}\right)\right]=O\left(\alpha_F\cdot Nn\epsilon\epsilon_0^{1-\beta}+\alpha_N\cdot Nn\epsilon_0^{2-\beta}\right).\tag{20}$$

To bound the KL-divergence conditioned on $P$, we start with the following bound

$$N\cdot\alpha_F\cdot\mathbb{E}_P\left[\log\left(\frac{V_{F,\sigma,t}+V_{N,\sigma,t}}{V_{F,1-\sigma,t}+V_{N,1-\sigma,t}}\right)\right]\leq t^*\cdot\mathbb{E}_P\left[\log\left(\frac{V_{F,\sigma,t}+V_{N,\sigma,t}}{V_{N,1-\sigma,t}}\right)\right]$$

$$=t^*\cdot\mathbb{E}_P\left[\log\left(\frac{V_{N,\sigma,t}}{V_{N,1-\sigma,t}}\right)+\log\left(\frac{V_{F,\sigma,t}+V_{N,\sigma,t}}{V_{N,\sigma,t}}\right)\right]$$

We have already calculated in (17) that

$$t^* \cdot \mathbb{E}_P \left[ \log \left( \frac{V_{N,\sigma,t}}{V_{N,1-\sigma,t}} \right) \right] \leq t^* n \epsilon \epsilon_0^{1-\beta}.$$

For the remaining term, note that

$$\mathbb{E}_P \left[ \log \left( \frac{V_{F,\sigma,t} + V_{N,\sigma,t}}{V_{N,\sigma,t}} \right) \right] = \mathbb{E}_P \left[ \log \left( \frac{V_{F,\sigma,t}}{V_{N,\sigma,t}} + 1 \right) \right] \leq \mathbb{E}_P \left[ \frac{V_{F,\sigma,t}}{V_{N,\sigma,t}} \right],$$

which can be calculated directly

$$
\begin{aligned}
\mathbb{E}_P \left[ \frac{V_{F,\sigma,t}}{V_{N,\sigma,t}} \right] &= \mathbb{E}_{\hat{n}_t} \left[ \mathbb{E}_{P|\hat{n}_t} \left[ \frac{V_{F,\sigma,t}}{V_{N,\sigma,t}} \bigg| \hat{n}_t \right] \right] \\
&= \frac{\alpha_F}{\alpha_N} \cdot \mathbb{E}_{P,\hat{n}_t} \left[ \sum_{s=0}^{\hat{n}_t} \left( \frac{p_{\sigma,x_1}}{q_{\sigma,x_1}} \right)^s \left( \frac{p_{1-\sigma,x_1}}{q_{1-\sigma,x_1}} \right)^{\hat{n}_t - s} \left( \frac{p_{1,x_0}}{q_{1,x_0}} \right)^{n-\hat{n}_t} \binom{\hat{n}_t}{s} \left( p_{\sigma|x_1} \right)^s \left( p_{1-\sigma|x_1} \right)^{\hat{n}_t - s} \right] \\
&= \frac{\alpha_F}{\alpha_N} \cdot \mathbb{E}_{P,\hat{n}_t} \left[ \left( \frac{p_{1,x_0}}{q_{1,x_0}} \right)^{n-\hat{n}_t} \sum_{s=0}^{\hat{n}_t} \binom{\hat{n}_t}{s} \left( \frac{p_{\sigma,x_1} p_{\sigma|x_1}}{q_{\sigma,x_1}} \right)^s \left( \frac{p_{1-\sigma,x_1} p_{1-\sigma|x_1}}{q_{1-\sigma,x_1}} \right)^{\hat{n}_t - s} \right] \\
&= \frac{\alpha_F}{\alpha_N} \cdot \mathbb{E}_{P,\hat{n}_t} \left[ \left( \frac{p_{1,x_0}}{q_{1,x_0}} \right)^{n-\hat{n}_t} \left( \frac{p_{\sigma,x_1} p_{\sigma|x_1}}{q_{\sigma,x_1}} + \frac{p_{1-\sigma,x_1} p_{1-\sigma|x_1}}{q_{1-\sigma,x_1}} \right)^{\hat{n}_t} \right] \\
&= \frac{\alpha_F}{\alpha_N} \cdot \sum_{s=0}^{n} \left( \frac{p_{1,x_0}}{q_{1,x_0}} \right)^{n-s} \left( \frac{p_{\sigma,x_1} p_{\sigma|x_1}}{q_{\sigma,x_1}} + \frac{p_{1-\sigma,x_1} p_{1-\sigma|x_1}}{q_{1-\sigma,x_1}} \right)^s \binom{n}{s} \left( p_{x_1} \right)^s \left( p_{x_0} \right)^{n-s} \\
&= \frac{\alpha_F}{\alpha_N} \cdot \left( \frac{p_{\sigma,x_1}^2}{q_{\sigma,x_1}} + \frac{p_{1-\sigma,x_1}^2}{q_{1-\sigma,x_1}} + \frac{p_{1,x_0}^2}{q_{1,x_0}} \right)^n.
\end{aligned}
$$

In the following, we analyze each summand separately:

$$\frac{p_{1,x_0}^2}{q_{1,x_0}} = \frac{\left( 1 - C_\beta \epsilon^\beta \right)^2}{1 - C_\beta \epsilon_0^\beta} = O(1),$$

and

$$\frac{p_{\sigma,x_1}^2}{q_{\sigma,x_1}} = \frac{\left( \frac{1}{2} C_\beta \epsilon^\beta + \frac{1}{2} \epsilon \right)^2}{\frac{1}{2} C_\beta \epsilon_0^\beta + \frac{1}{2} \epsilon_0} = O \left( \frac{\epsilon^{2\beta}}{\epsilon_0^\beta} \right), \quad \frac{p_{1-\sigma,x_1}^2}{q_{1-\sigma,x_1}} = \frac{\left( \frac{1}{2} C_\beta \epsilon^\beta - \frac{1}{2} \epsilon \right)^2}{\frac{1}{2} C_\beta \epsilon_0^\beta - \frac{1}{2} \epsilon_0} = O \left( \frac{\epsilon^{2\beta}}{\epsilon_0^\beta} \right).$$

When $\epsilon = (n\sqrt{N})^{-1/(2-\beta)}$ and $\epsilon_0 = (nN)^{-1/(2-\beta)}$, we have

$$t^* \cdot \mathbb{E}_P \left[ \frac{V_{F,\sigma,t}}{V_{N,\sigma,t}} \right] = O \left( t^* \cdot \frac{\alpha_F}{\alpha_N} \cdot \left( \frac{\epsilon^{2\beta}}{\epsilon_0^\beta} \right)^n \right) = O \left( \frac{(t^*)^2}{N} \cdot \left( \frac{1}{n} \right)^{n\beta/(2-\beta)} \right),$$

and also the other terms are of order

$$O \left( \alpha_F \cdot N n \epsilon \epsilon_0^{1-\beta} + \alpha_N \cdot N n \epsilon_0^{2-\beta} \right) = O \left( \frac{t^*}{N^{(3/2-\beta)/(2-\beta)}} + 1 \right).$$

Altogether, we have

$$\mathbb{D}_{\mathrm{KL}}(P_\sigma || P_{1-\sigma}) = O \left( \frac{t^*}{N^{(3/2-\beta)/(2-\beta)}} + 1 + \frac{(t^*)^2}{N} \cdot \left( \frac{1}{n} \right)^{n\beta/(2-\beta)} \right).$$

When

$$t^* \leq \sqrt{N \cdot n^{n\beta/(2-\beta)}} \quad \text{and} \quad N \geq n^{n\beta/(1-\beta)},$$

we have $\mathbb{D}_{\mathrm{KL}}(P_\sigma || P_{1-\sigma}) = O(1)$, and thus by Lemma E.6, we can guarantee ① as

$$\mathbb{P}_{\boldsymbol{I}^*}\left(\mathbb{P}_Z\left(\psi(Z) \neq \sigma | \boldsymbol{I}^*\right) \geq \frac{2 - \sqrt{2}}{8}\right) \geq \frac{2 - \sqrt{2}}{8}.$$

Moreover, ② follows from the multiplicative Chernoff bound (Lemma E.2). Since $|\boldsymbol{I}^*| \sim \mathrm{Bin}(N, \alpha_F)$,

$$\mathbb{P}_{\boldsymbol{I}^*}\left(|\boldsymbol{I}^*| \geq \frac{t^*}{2}\right) = 1 - \mathbb{P}_{\boldsymbol{I}^*}\left(|\boldsymbol{I}^*| < \frac{t^*}{2}\right) \geq 1 - \exp\left(-\frac{t^*}{8}\right).$$

The proof is done! $\qquad\qquad\square$

## D. Additional Results

In this section, we make an initial trial towards understanding the optimal adaptivity. Inspired by the example in Subsection 5.3, we focus on exploring under what multitask situations (what assumptions on multitask hyperparameters) pooling achieves optimal adaptive rates, especially when pooling is not minimax optimal. We start with the simple case of $\beta = 0$. Specifically, we build a lower bound by constructing a multitask problem given a set of transfer exponents, and show that no adaptive algorithm can achieve a learning rate better than pooling, which however is not minimax optimal. Future works could be to focus on simplifying the assumptions therein (hopefully only having neat assumptions on the set of transfer exponents), which could be hard as it requires a clean and tight bound on the KL divergence between mixture distributions.

**Theorem D.1.** *Suppose that $|\mathcal{H}| \geq 3$. Given a set of transfer exponents $\{\rho_t, t \in [N+1]\}$ with every $\rho_t \geq 1$. Let $\rho_{(1)} \leq \cdots \leq \rho_{(N+1)}$ be transfer exponents in order. Assume that each dataset is of size $n$. Let $t_p^* = \arg\min_{t \in [N+1]}((nt^2)/N)^{-1/(2\bar\rho_t)}$ be the optimal cut-off in the pooling bound and $\epsilon_p^* = (n(t_p^*)^2/N)^{-1/2\rho_{(t_p^*)}}$ be the pooling rates. If there exists a positive integer $N_G \leq N$ such that the following hold*

$$N_G = O\left(\frac{1}{n(\epsilon_p^*)^2 e^{n(\epsilon_p^*)^2}}\right) \quad \text{and} \quad \rho_{(s)} \geq \frac{2\rho_{(t_p^*)} \ln(nN)}{\ln(n(t_p^*)^2/N)}, \quad \forall s \in \{N_G + 1, \ldots, N\},$$

*then for any adaptive algorithm $\mathcal{A}$, there is a multitask model $(\mathcal{M}, \Pi)$ satisfying a Bernstein class condition with $\beta = 0$ such that, given a multisample $Z \sim \Pi$, we have*

$$\mathbb{E}\left[\mathcal{E}_\mathcal{D}(\mathcal{A}(Z))\right] = \Omega\left(\epsilon_p^*\right).$$

*Proof of Theorem D.1.* Since $|\mathcal{H}| \geq 3$, we can find two points $x_0, x_1 \in \mathcal{X}$ such that the following holds. There exists $y_0 \in \{0, 1\}$ such that there are two concepts $h_0, h_1 \in \mathcal{H}$ satisfying $h_0(x_0) = h_1(x_0) = y_0$, $h_0(x_1) = 0$ and $h_1(x_1) = 1$. Given a multiset of transfer exponents $\{\rho_t, 1 \leq t \leq N + 1\}$, for each $t \in [N + 1]$, we construct two distributions as follow. Fix any $\epsilon \in (0, 1)$. For each $\sigma \in \{0, 1\}$, let

$$P_t^\sigma : \ P_t^\sigma(x_1) = 1, \ \ P_t^\sigma(Y = 1 | X = x_1) = 1/2 + (1/2)(2\sigma - 1)\epsilon^{\rho_t}.$$

It is clear that all the distributions $\{P_t^\sigma, 1 \leq t \leq N + 1\}$ share the same optimal function $h_\sigma$. Next, we verify that every $P_t^\sigma$ satisfy a Bernstein class condition with $\beta = 0$ and some $C_\beta \geq 2$. This is trivial by definition: $P_t^\sigma(h_{1-\sigma} \neq h_\sigma) = 1 \leq C_\beta = C_\beta\left(\mathcal{E}_{P_t^\sigma}(h_{1-\sigma})\right)^\beta$. Finally, we have to verify that the constructed distributions admit the given set of transfer exponents. Let $P_{N+1}^\sigma$ be the target distribution $\mathcal{D}^\sigma$ and thus $\rho_{N+1} = 1$. Note that for each $t \in [N]$ and any $h \in \mathcal{H}$, we have

$$\mathcal{E}_{P_t^\sigma}(h) = \mathrm{er}_{P_t^\sigma}(h) - \mathrm{er}_{P_t^\sigma}(h_\sigma) = \epsilon^{\rho_t} \mathbb{1}\left\{h(x_1) \neq \sigma\right\}.$$

Therefore, we have for each $t \in [N]$ and any $h \in \mathcal{H}$ that

$$\mathcal{E}_{\mathcal{D}^\sigma}(h) = \epsilon \mathbb{1}\left\{h(x_1) \neq \sigma\right\} = \left[(\epsilon \mathbb{1}\left\{h(x_1) \neq \sigma\right\})^{\rho_t}\right]^{1/\rho_t} = \left[\epsilon^{\rho_t} \mathbb{1}\left\{h(x_1) \neq \sigma\right\}\right]^{1/\rho_t} \leq C_\rho \left(\mathcal{E}_{P_t^\sigma}(h)\right)^{1/\rho_t}$$

with some constant $C_\rho \geq 2$. The following lemma gives a probabilistic method for constructing an i.i.d. sequence of distributions admit the given transfer exponents with some descent probability.

**Lemma D.2.** *Let $\{\rho_t, t = 1, \ldots, N + 1\}$ be a multiset of transfer exponents (to some target $\mathcal{D}$). Given a set of distributions $P_1, \ldots, P_{N+1}$ such that $P_t$ admits $\rho_t$ for every $t$. Let $P_{(1)}, \ldots, P_{(N+1)}$ denote the ordered sequence of $P_1, \ldots, P_{N+1}$ based on a non-decreasing order of their transfer exponents $\rho_{(1)} \leq \ldots \leq \rho_{(N+1)}$. Consider the following random construction of distributions: for every $t \in [N + 1]$, draw $i_t \sim \text{Unif}(\{1, \ldots, \lfloor(N + 1)/10\rfloor\})$ and set $Q_t = P_{(i_t)}$. Then, with probability at least $0.97$, there exists a permutation of $Q_1, \ldots, Q_{N+1}$ that admits $\rho_1, \ldots, \rho_{N+1}$.*

*Proof of Lemma D.2.* Note that for any $t \in [N + 1]$, if $Q_t = P_{(i_t)}$ satisfies $i_t \leq t$ and thus $\rho_{(i_t)} \leq \rho_{(t)}$, then it holds

$$\mathcal{E}_{\mathcal{D}}(h) \leq C_\rho(\mathcal{E}_{P_{(i_t)}}(h))^{1/\rho_{(i_t)}} \leq C_\rho(\mathcal{E}_{P_{(i_t)}}(h))^{1/\rho_{(t)}} = C_\rho(\mathcal{E}_{Q_{(t)}}(h))^{1/\rho_{(t)}},$$

namely, $Q_{(t)}$ admits a transfer exponent $\rho_{(t)}$. Accordingly, it suffices to show that with some descent probability, $\{i_1, \ldots, i_{N+1}\}$ has a permutation $\{i'_1, \ldots, i'_{N+1}\}$ such that $i'_t \leq t$ for every $t \in [N + 1]$.

Since $i_t \sim \text{Unif}(\{1, \ldots, \lfloor(N + 1)/10\rfloor\})$, this is always true for $t > \lfloor(N + 1)/10\rfloor$. For notational simplicity, let $m = \lfloor(N+1)/10\rfloor$ and the problem is to draw $10m$ i.i.d. samples $\{T_1, \ldots, T_{10m}\}$ from $\text{Unif}([m])$. Let $S_i = \sum_{j=1}^{10m} \mathbb{1}\{T_j \leq i\}$ denote the number of random samples with value at most $i$. It is clear that $S_i \sim \text{Bin}(10m, i/m)$. Note that if $S_1 \geq 1, \ldots, S_m \geq m$, then we are able to greedily find $m$ samples $\{T_1, \ldots, T_m\}$ such that $T_i \leq i$ for every $i \in [m]$. Hence, we have

$$\mathbb{P}\left(\exists T_1, \ldots, T_m \in \{T_1, \ldots, T_{10m}\} \text{ s.t. } T_i \leq i, \forall i \in [m]\right) \geq \mathbb{P}\left(S_1 \geq 1, \ldots, S_m \geq m\right).$$

To lower bound the probability on the above RHS, we consider the complementary event. By union bound, we have $\mathbb{P}(\exists i \in [m] : S_i < i) \leq \sum_{i=1}^m \mathbb{P}(S_i < i)$. Since $S_i \sim \text{Bin}(10m, i/m)$, for each $i \in [m]$, by multiplicative Chernoff bound (Lemma E.2 with $\delta = 9/10$), we have that

$$\mathbb{P}(S_i < i) \leq \exp\left(-81i/20\right).$$

It follows that

$$\mathbb{P}(\exists i \in [m] : S_i < i) \leq \sum_{i=1}^m \mathbb{P}(S_i < i) \leq \sum_{i=1}^m \exp\left(-81i/20\right) \leq \sum_{i=1}^{\infty} \exp\left(-81i/20\right) = \frac{e^{-81/20}}{1 - e^{-81/20}} \leq 0.03.$$

Altogether, we have with probability of at least $1 - 0.03 = 0.97$, there exists a permutation of $Q_1, \ldots, Q_{N+1}$ admits $\rho_1, \ldots, \rho_{N+1}$. $\square$

We apply Lemma D.2 to the distributions $P_1^\sigma, \ldots, P_{N+1}^\sigma$ and get independently random distributions $Q_1^\sigma, \ldots, Q_{N+1}^\sigma$ that admit $\rho_1, \ldots, \rho_{N+1}$ with probability at least $0.97$. Now, for each $\sigma \in \{0, 1\}$, assume that a multisample $Z \sim \Pi^\sigma$ consists of $n_t$ samples drawn from each distribution $Q_t^\sigma$. We denote by $\Pi^\sigma = \Pi_{t=1}^{N+1} Q_t^\sigma$ the (joint) multitask distribution. For the same purpose as in the proof of Theorem 5.2, making such a random construction can facilitate the analysis of the KL-divergence between $\Pi^\sigma$ and $\Pi^{1-\sigma}$ due to the independence between datasets. For notational simplicity, denote $m = \lfloor(N + 1)/10\rfloor$. Based on the chain rule for KL-divergence, i.e., $\mathbb{D}_{\text{KL}}(P(X, Y)\|Q(X, Y)) = \mathbb{D}_{\text{KL}}(P(X)\|Q(X)) + \mathbb{D}_{\text{KL}}(P(Y|X)\|Q(Y|X))$, we can write

$$\mathbb{D}_{\text{KL}}\left(\Pi^1\|\Pi^0\right) = \sum_{t=1}^{N+1} \mathbb{D}_{\text{KL}}\left(Q_t^1\|Q_t^0\right) = \sum_{t=1}^{N+1} \mathbb{D}_{\text{KL}}\left(\frac{1}{m}\sum_{s=1}^m P_{(s)}^1 \bigg\| \frac{1}{m}\sum_{s=1}^m P_{(s)}^0\right).$$

For simplicity, we assume that $n_t = n$ for every $t \in [N + 1]$. Now given a dataset $Z_t$ with size $n$, we consider a sufficient statistic $\hat{n}_t$ representing the number of $(x_1, 1)$ examples in $Z_t$. The likelihood function of $\hat{n}_t$ on the mixture distribution $\frac{1}{m}\sum_{s=1}^m P_{(s)}^1$ is

$$\left(\frac{1}{m}\sum_{s=1}^m P_{(s)}^1\right)(\hat{n}_t) = \frac{1}{m}\sum_{s=1}^m \binom{n}{\hat{n}_t}\left(\frac{1}{2} + \frac{1}{2}\epsilon^{\rho(s)}\right)^{\hat{n}_t}\left(\frac{1}{2} - \frac{1}{2}\epsilon^{\rho(s)}\right)^{n-\hat{n}_t},$$

and similarly, we can write out the likelihood of $\hat{n}_t$ on $\frac{1}{m}\sum_{s=1}^{m}P_{(s)}^0$. It follows that

$$
\begin{aligned}
\mathbb{D}_{\mathrm{KL}}\left(\Pi^1 \| \Pi^0\right) &= \sum_{t=1}^{N+1} \mathbb{D}_{\mathrm{KL}}\left(\left(\frac{1}{m}\sum_{s=1}^{m}P_{(s)}^1\right)(\hat{n}_t) \,\middle\|\, \left(\frac{1}{m}\sum_{s=1}^{m}P_{(s)}^0\right)(\hat{n}_t)\right) \\
&= \sum_{t=1}^{N+1} \frac{1}{m}\sum_{k=1}^{m} \mathbb{E}_{\hat{n}_t \sim P_{(k)}^1}\left[\log\left(\frac{\sum_{s=1}^{m}P_{(s)}^1(\hat{n}_t)}{\sum_{s=1}^{m}P_{(s)}^0(\hat{n}_t)}\right)\right] \\
&= \sum_{t=1}^{N+1} \frac{1}{m}\sum_{k=1}^{m} \mathbb{E}_{\hat{n}_t \sim \mathrm{Bin}(n,1/2+\epsilon^{\rho(k)}/2)}\left[\log\left(\frac{\sum_{s=1}^{m}\left(1/2+\epsilon^{\rho(s)}/2\right)^{\hat{n}_t}\left(1/2-\epsilon^{\rho(s)}/2\right)^{n-\hat{n}_t}}{\sum_{s=1}^{m}\left(1/2-\epsilon^{\rho(s)}/2\right)^{\hat{n}_t}\left(1/2+\epsilon^{\rho(s)}/2\right)^{n-\hat{n}_t}}\right)\right] \\
&= \sum_{t=1}^{N+1} \frac{1}{m}\sum_{k=1}^{m} \mathbb{E}_{\hat{n}_t \sim \mathrm{Bin}(n,1/2+\epsilon^{\rho(k)}/2)}\left[\log\left(\frac{\sum_{s=1}^{m}\left(1+\epsilon^{\rho(s)}\right)^{\hat{n}_t}\left(1-\epsilon^{\rho(s)}\right)^{n-\hat{n}_t}}{\sum_{s=1}^{m}\left(1-\epsilon^{\rho(s)}\right)^{\hat{n}_t}\left(1+\epsilon^{\rho(s)}\right)^{n-\hat{n}_t}}\right)\right].
\end{aligned}
$$

For any $s \in [m]$, assume that $\hat{n}_t \leq n/2 + \Delta$ for some $\Delta > 0$, then we have

$$
\begin{aligned}
\frac{\left(1+\epsilon^{\rho(s)}\right)^{\hat{n}_t}\left(1-\epsilon^{\rho(s)}\right)^{n-\hat{n}_t}}{\left(1-\epsilon^{\rho(s)}\right)^{\hat{n}_t}\left(1+\epsilon^{\rho(s)}\right)^{n-\hat{n}_t}} &\leq \frac{\left(1+\epsilon^{\rho(s)}\right)^{n/2+\Delta}\left(1-\epsilon^{\rho(s)}\right)^{n/2-\Delta}}{\left(1-\epsilon^{\rho(s)}\right)^{n/2+\Delta}\left(1+\epsilon^{\rho(s)}\right)^{n/2-\Delta}} \\
&= \left(\frac{1+\epsilon^{\rho(s)}}{1-\epsilon^{\rho(s)}}\right)^{2\Delta} = \left(1+\frac{2\epsilon^{\rho(s)}}{1-\epsilon^{\rho(s)}}\right)^{2\Delta} \leq \exp\left\{\frac{4\Delta\epsilon^{\rho(s)}}{1-\epsilon^{\rho(s)}}\right\},
\end{aligned}
$$

where the last inequality uses the fact that $(1+x)^n \leq e^{nx}$ for any $x > -1$. Choosing $\Delta = (8N\epsilon^{\rho(s)})^{-1}$ will guarantee that

$$
\frac{\left(1+\epsilon^{\rho(s)}\right)^{\hat{n}_t}\left(1-\epsilon^{\rho(s)}\right)^{n-\hat{n}_t}}{\left(1-\epsilon^{\rho(s)}\right)^{\hat{n}_t}\left(1+\epsilon^{\rho(s)}\right)^{n-\hat{n}_t}} \leq \exp\left\{\frac{4\Delta\epsilon^{\rho(s)}}{1-\epsilon^{\rho(s)}}\right\} \leq e^{1/N}.
$$

Next, we quantify the probability of the event that $\hat{n}_t \geq \frac{n}{2} + \Delta$. For any fixed $\Delta > 0$, a straightforward observation is that for any $k \in [m]$, since $\rho_{(k)} \geq 1$, we have

$$
\mathbb{P}_{\hat{n}_t \sim \mathrm{Bin}(n,1/2+\epsilon^{\rho(k)}/2)}\left(\hat{n}_t \geq \frac{n}{2} + \Delta\right) \leq \mathbb{P}_{\hat{n}_t \sim \mathrm{Bin}(n,1/2+\epsilon/2)}\left(\hat{n}_t \geq \frac{n}{2} + \Delta\right). \tag{21}
$$

**Lemma D.3** (Chernoff-Hoeffding theorem). *Let $X \sim \mathrm{Bin}(m,p)$. For any $x > 0$, we have*

$$
\mathbb{P}\left(\frac{X}{m} \geq p + x\right) \leq \exp\left\{-\frac{x^2 m}{2p(1-p)}\right\}.
$$

Now for any $k \in [m]$, by the above Lemma D.3, we have

$$
\mathbb{P}_{\hat{n}_t \sim \mathrm{Bin}(n,1/2+\epsilon/2)}\left(\hat{n}_t \geq \frac{n}{2} + \Delta\right) = \mathbb{P}\left(\frac{\hat{n}_t}{n} \geq \frac{1}{2} + \frac{\Delta}{n}\right) \leq \exp\left\{-\frac{n\left(\frac{\Delta}{n} - \frac{\epsilon}{2}\right)^2}{2\left(\frac{1}{2}+\frac{\epsilon}{2}\right)\left(\frac{1}{2}-\frac{\epsilon}{2}\right)}\right\}.
$$

Plugging $\Delta = (8N\epsilon^{\rho(s)})^{-1}$ into the RHS above, we can further derive

$$
\mathbb{P}_{\hat{n}_t \sim \mathrm{Bin}(n,1/2+\epsilon/2)}\left(\hat{n}_t \geq \frac{n}{2} + \Delta\right) \leq \exp\left\{-2n\left(\frac{1}{8nN\epsilon^{\rho(s)}} - \frac{\epsilon}{2}\right)^2\right\}. \tag{22}
$$

Intuitively, for a sufficiently large index $s$ (such that its transfer exponent $\rho_{(s)}$ is sufficiently large), the above upper bound on the probability can be small. Concretely, we choose $\epsilon$ to be the following pooling rate (obtained from Lemma 6.1 with $\beta = 0$), i.e.,

$$
\epsilon = \min_{t \in [N+1]}\left(c_1 \cdot \frac{N}{nt^2}\right)^{1/2\rho(t)} =: \min_{t \in [N+1]}\left(c_1 \cdot \phi_{(t)}\right)^{1/2\rho(t)}
$$

with some numerical constant $c_1 > 0$. Since we always have $\bar{\rho}_t \leq \rho_{(t)}$, proving a lower bound of $\epsilon$ would directly implies a lower bound of the pooling rates. Let $t_p^* \in [N+1]$ be the optimal cut-off in the above minimum expression, which can be different from the optimal cut-off $t^*$ in the minimax rates. Then, for any $s \in [N+1]$, we have

$$
\left(c_1 \cdot \phi_{(s)}\right)^{1/2\rho(s)} \geq \left(c_1 \cdot \phi_{(t_p^*)}\right)^{1/2\rho(t_p^*)} \quad\Rightarrow\quad \epsilon^{2\rho(s)} = \left(c_1 \cdot \phi_{(t_p^*)}\right)^{\rho(s)/\rho(t_p^*)} \leq c_1 \cdot \phi_{(s)} = \frac{c_1 N}{ns^2}.
$$

*Remark* D.4. We underline the following observations:

(1) By definition, the minimax rate is $(\frac{1}{nt^*})^{1/2\rho_{(t^*)}}$ and the pooling rate satisfies

$$\min_{t\in[N+1]} \left(\frac{N}{nt^2}\right)^{1/2\rho_{(t)}} \leq \left(\frac{N}{n(t^*)^2}\right)^{1/2\rho_{(t^*)}}.$$

Hence, if $t^* = \Omega(N)$, pooling achieves the minimax rates and is thus optimal.

(2) If $t_p^* \leq \sqrt{N/n}$, then the pooling rate $(\frac{N}{n(t_p^*)^2})^{1/2\rho_{(t_p^*)}}$ is vacuous.

Therefore, we can assume that $t^* = o(N)$ and $t_p^* \geq \sqrt{N/n}$.

Note that

$$\frac{\epsilon}{2} = O\left(\left(\frac{N}{n(t_p^*)^2}\right)^{1/2\rho_{(t_p^*)}}\right) \quad \text{and} \quad \frac{1}{8nN\epsilon^{\rho_{(s)}}} = \Omega\left(\frac{1}{nN}\left(\frac{n(t_p^*)^2}{N}\right)^{\rho_{(s)}/2\rho_{(t_p^*)}}\right).$$

For those large enough indices $s$,

$$\rho_{(s)} \geq \frac{2\rho_{(t_p^*)}\ln(nN)}{\ln(n(t_p^*)^2/N)} \quad \Rightarrow \quad \begin{cases} \rho_{(s)} \geq \frac{2\rho_{(t_p^*)}\ln(nN)}{\ln(n(t_p^*)^2/N)} - 1 \quad \Rightarrow \quad \epsilon = O((nN\epsilon^{\rho_{(s)}})^{-1}) \\ \rho_{(s)} \geq \frac{2\rho_{(t_p^*)}\ln(nN)}{\ln(n(t_p^*)^2/N)} \quad \Rightarrow \quad (nN\epsilon^{\rho_{(s)}})^{-1} = \Omega(1) \end{cases}, \tag{23}$$

Putting (21), (22) and (23) together, we can bound for any $k \in [m]$ that

$$\mathbb{P}_{\hat{n}_t\sim\text{Bin}(n,1/2+\epsilon^{\rho_{(k)}}/2)}\left(\hat{n}_t \geq \frac{n}{2} + \Delta\right) = O\left(\exp\left\{-n\left(\frac{1}{nN\epsilon^{\rho_{(s)}}}\right)^2\right\}\right) = O\left(e^{-n}\right).$$

When $\hat{n}_t \geq \frac{n}{2} + \Delta$, we can upper bound it by the following extreme case

$$\frac{(1+\epsilon^{\rho_{(s)}})^{\hat{n}_t}(1-\epsilon^{\rho_{(s)}})^{n-\hat{n}_t}}{(1-\epsilon^{\rho_{(s)}})^{\hat{n}_t}(1+\epsilon^{\rho_{(s)}})^{n-\hat{n}_t}} \leq \left(\frac{1+\epsilon^{\rho_{(s)}}}{1-\epsilon^{\rho_{(s)}}}\right)^n \leq e^{4n\epsilon^{\rho_{(s)}}}.$$

For those small indices $s \in [m]$ such that (23) fails, we have

$$\frac{(1+\epsilon^{\rho_{(s)}})^{\hat{n}_t}(1-\epsilon^{\rho_{(s)}})^{n-\hat{n}_t}}{(1-\epsilon^{\rho_{(s)}})^{\hat{n}_t}(1+\epsilon^{\rho_{(s)}})^{n-\hat{n}_t}} \leq \frac{(1+\epsilon)^{\hat{n}_t}(1-\epsilon)^{n-\hat{n}_t}}{(1-\epsilon)^{\hat{n}_t}(1+\epsilon)^{n-\hat{n}_t}} = \left(\frac{1+\epsilon}{1-\epsilon}\right)^{2\hat{n}_t-n} \leq e^{8\epsilon(\hat{n}_t-n/2)}.$$

For notational simplicity, we denote by $\mathcal{S}$ the set of those indices $s \in [m]$ satisfying (23) and denote $\mathcal{S}^c := [m] \setminus \mathcal{S}$. We further define

$$\hat{\alpha}_{\mathcal{S}} := \frac{\sum_{s\in\mathcal{S}}(1+\epsilon^{\rho_{(s)}})^{\hat{n}_t}(1-\epsilon^{\rho_{(s)}})^{n-\hat{n}_t}}{\sum_{s\in[m]}(1+\epsilon^{\rho_{(s)}})^{\hat{n}_t}(1-\epsilon^{\rho_{(s)}})^{n-\hat{n}_t}} \quad \text{and} \quad \hat{\alpha}_{\mathcal{S}^c} := \frac{\sum_{s\in\mathcal{S}^c}(1+\epsilon^{\rho_{(s)}})^{\hat{n}_t}(1-\epsilon^{\rho_{(s)}})^{n-\hat{n}_t}}{\sum_{s\in[m]}(1+\epsilon^{\rho_{(s)}})^{\hat{n}_t}(1-\epsilon^{\rho_{(s)}})^{n-\hat{n}_t}}.$$

Now for any $k \in [m]$, by applying Lemma E.5, we can bound

$$\mathbb{E}_{\hat{n}_t\sim\text{Bin}(n,1/2+\epsilon^{\rho_{(k)}}/2)}\left[\log\left(\frac{\sum_{s=1}^m(1+\epsilon^{\rho_{(s)}})^{\hat{n}_t}(1-\epsilon^{\rho_{(s)}})^{n-\hat{n}_t}}{\sum_{s=1}^m(1-\epsilon^{\rho_{(s)}})^{\hat{n}_t}(1+\epsilon^{\rho_{(s)}})^{n-\hat{n}_t}}\right)\right]$$

$$= \mathbb{E}_{\hat{n}_t\sim\text{Bin}(n,1/2+\epsilon^{\rho_{(k)}}/2)}\left[\log\left(\frac{\sum_{s\in\mathcal{S}}(1+\epsilon^{\rho_{(s)}})^{\hat{n}_t}(1-\epsilon^{\rho_{(s)}})^{n-\hat{n}_t} + \sum_{s\in\mathcal{S}^c}(1+\epsilon^{\rho_{(s)}})^{\hat{n}_t}(1-\epsilon^{\rho_{(s)}})^{n-\hat{n}_t}}{\sum_{s\in\mathcal{S}}(1-\epsilon^{\rho_{(s)}})^{\hat{n}_t}(1+\epsilon^{\rho_{(s)}})^{n-\hat{n}_t} + \sum_{s\in\mathcal{S}^c}(1-\epsilon^{\rho_{(s)}})^{\hat{n}_t}(1+\epsilon^{\rho_{(s)}})^{n-\hat{n}_t}}\right)\right]$$

$$\leq \mathbb{E}_{\hat{n}_t\sim\text{Bin}(n,1/2+\epsilon^{\rho_{(k)}}/2)}\left[\hat{\alpha}_{\mathcal{S}}\log\left(\frac{\sum_{s\in\mathcal{S}}(1+\epsilon^{\rho_{(s)}})^{\hat{n}_t}(1-\epsilon^{\rho_{(s)}})^{n-\hat{n}_t}}{\sum_{s\in\mathcal{S}}(1-\epsilon^{\rho_{(s)}})^{\hat{n}_t}(1+\epsilon^{\rho_{(s)}})^{n-\hat{n}_t}}\right)\right]$$

$$+ \mathbb{E}_{\hat{n}_t\sim\text{Bin}(n,1/2+\epsilon^{\rho_{(k)}}/2)}\left[\hat{\alpha}_{\mathcal{S}^c}\log\left(\frac{\sum_{s\in\mathcal{S}^c}(1+\epsilon^{\rho_{(s)}})^{\hat{n}_t}(1-\epsilon^{\rho_{(s)}})^{n-\hat{n}_t}}{\sum_{s\in\mathcal{S}^c}(1-\epsilon^{\rho_{(s)}})^{\hat{n}_t}(1+\epsilon^{\rho_{(s)}})^{n-\hat{n}_t}}\right)\right]$$

$$\leq \frac{1}{N} + \max_{s\in\mathcal{S}}\{4n\epsilon^{\rho_{(s)}}\}\cdot O\left(e^{-n}\right) + 8\epsilon\cdot\mathbb{E}_{\hat{n}_t\sim\text{Bin}(n,1/2+\epsilon^{\rho_{(k)}}/2)}\left[\hat{\alpha}_{\mathcal{S}^c}\left(\hat{n}_t - \frac{n}{2}\right)\right]. \tag{24}$$

When $s \in \mathcal{S}$, i.e., when (23) holds, it is clear that $\max_{s \in \mathcal{S}}\{n\epsilon^{\rho(s)}\} \cdot O(e^{-n}) = O(e^{-n}/N) = O(1/N)$. It remains to bound the last summand in (24). When $\hat{n}_t \leq n/2$, the last summand is non-positive and thus upper bounded by zero. Hence, we consider when $\hat{n}_t \geq n/2$. Note that

$$\hat{\alpha}_{\mathcal{S}^c} = \frac{\sum_{s \in \mathcal{S}^c}(1 + \epsilon^{\rho(s)})^{\hat{n}_t}(1 - \epsilon^{\rho(s)})^{n - \hat{n}_t}}{\sum_{s \in [m]}(1 + \epsilon^{\rho(s)})^{\hat{n}_t}(1 - \epsilon^{\rho(s)})^{n - \hat{n}_t}} \leq \frac{\sum_{s \in \mathcal{S}^c}(1 + \epsilon^{\rho(s)})^{\hat{n}_t}(1 - \epsilon^{\rho(s)})^{n - \hat{n}_t}}{\sum_{s \in \mathcal{S}}(1 + \epsilon^{\rho(s)})^{\hat{n}_t}(1 - \epsilon^{\rho(s)})^{n - \hat{n}_t}}.$$

When $\hat{n}_t \geq n/2$, for any $s \in \mathcal{S}^c$,

$$(1 + \epsilon^{\rho(s)})^{\hat{n}_t}(1 - \epsilon^{\rho(s)})^{n - \hat{n}_t} \leq (1 + \epsilon)^{\hat{n}_t}(1 - \epsilon)^{n - \hat{n}_t},$$

and for any $s \in \mathcal{S}$, $(1 + \epsilon^{\rho(s)})^{\hat{n}_t}(1 - \epsilon^{\rho(s)})^{n - \hat{n}_t} \geq 1$. It follows that

$$\hat{\alpha}_{\mathcal{S}^c} \leq \frac{|\mathcal{S}^c|}{|\mathcal{S}|}(1 + \epsilon)^{\hat{n}_t}(1 - \epsilon)^{n - \hat{n}_t} = \frac{|\mathcal{S}^c|}{|\mathcal{S}|}(1 - \epsilon^2)^{n/2}\left(\frac{1 + \epsilon}{1 - \epsilon}\right)^{\hat{n}_t - n/2} \leq \frac{|\mathcal{S}^c|e^{n\epsilon^2/2 + 4\epsilon(\hat{n}_t - n/2)}}{|\mathcal{S}|},$$

and thus for any $k \in [m]$,

$$\mathbb{E}_{\hat{n}_t \sim \mathrm{Bin}(n, 1/2 + \epsilon^{\rho(k)}/2)}\left[\hat{\alpha}_{\mathcal{S}^c}\left(\hat{n}_t - \frac{n}{2}\right)\right] \leq \frac{|\mathcal{S}^c|e^{n\epsilon^2/2}}{|\mathcal{S}|} \cdot \mathbb{E}_{\hat{n}_t \sim \mathrm{Bin}(n, 1/2 + \epsilon^{\rho(k)}/2)}\left[\left(\hat{n}_t - \frac{n}{2}\right)e^{4\epsilon(\hat{n}_t - n/2)}\right]$$

$$= O\left(\frac{|\mathcal{S}^c|n\epsilon e^{n\epsilon^2}}{N}\right).$$

Recall that the Moment Generating Function (MGF) of a binomial distribution $Z \sim \mathrm{Bin}(n, p)$ is $M_Z(x) = (1 - p + pe^x)^n$. Since $\hat{n}_t \sim \mathrm{Bin}(n, 1/2 + \epsilon^{\rho(k)}/2)$, the MGF of $\hat{n}_t$ is

$$M_{\hat{n}_t}(x) = \left(\frac{1}{2} - \frac{\epsilon^{\rho(k)}}{2} + \left(\frac{1}{2} + \frac{\epsilon^{\rho(k)}}{2}\right)e^x\right)^n.$$

Let $\hat{\mu}_t := \hat{n}_t - n/2$. By definition, we have

$$M_{\hat{\mu}_t}(x) = \mathbb{E}\left[e^{x\hat{\mu}_t}\right] = \mathbb{E}\left[e^{x(\hat{n}_t - n/2)}\right] = e^{-nx/2} \cdot \mathbb{E}\left[e^{x\hat{n}_t}\right] = e^{-nx/2} \cdot M_{\hat{n}_t}(x).$$

It follows that

$$M_{\hat{\mu}_t}(x) = e^{-nx/2} \cdot M_{\hat{n}_t}(x) = e^{-nx/2} \cdot \left(\frac{1}{2} - \frac{\epsilon^{\rho(k)}}{2} + \left(\frac{1}{2} + \frac{\epsilon^{\rho(k)}}{2}\right)e^x\right)^n.$$

The purpose of calculating this MGF is to utilize the fact that $M'_{\hat{\mu}_t}(x) = \mathbb{E}[\hat{\mu}_t e^{x\hat{\mu}_t}]$. In particular, choosing $x = 4\epsilon$ yields

$$\mathbb{E}_{\hat{n}_t \sim \mathrm{Bin}(n, 1/2 + \epsilon^{\rho(k)}/2)}\left[\left(\hat{n}_t - \frac{n}{2}\right)e^{4\epsilon(\hat{n}_t - n/2)}\right] = \mathbb{E}\left[\hat{\mu}_t e^{4\epsilon\hat{\mu}_t}\right] = M'_{\hat{\mu}_t}(4\epsilon)$$

$$= ne^{-2n\epsilon}\left(\frac{e^{4\epsilon} + 1}{2} + \frac{(e^{4\epsilon} - 1)\epsilon^{\rho(k)}}{2}\right)^{n-1}\left(\frac{e^{4\epsilon} - 1}{4} + \frac{(e^{4\epsilon} + 1)\epsilon^{\rho(k)}}{4}\right)$$

$$\leq ne^{-2n\epsilon}\left(\frac{e^{4\epsilon} + 1}{2} + \frac{(e^{4\epsilon} - 1)\epsilon}{2}\right)^{n-1}\left(\frac{e^{4\epsilon} - 1}{4} + \frac{(e^{4\epsilon} + 1)\epsilon}{4}\right)$$

$$\leq \frac{ne^{4\epsilon - 2n\epsilon}}{2}\left(\frac{e^{4\epsilon} + 1}{2} + \frac{(e^{4\epsilon} - 1)\epsilon}{2}\right)^{n-1}$$

Therefore, the following assumption would suffice to guarantee that $\epsilon \cdot \mathbb{E}[\hat{\alpha}_{\mathcal{S}^c}(\hat{n}_t - n/2)] = O(1/N)$

$$|\mathcal{S}^c| = O\left(\frac{1}{n\epsilon^2 e^{n\epsilon^2}}\right). \tag{25}$$

Altogether, we have guaranteed that $\mathbb{D}_{\mathrm{KL}}\left(\Pi^1 || \Pi^0\right) = O(1)$ following from (24). This implies that no adaptive algorithm can outperform pooling. $\qquad\square$

# E. Technical lemmas

**Lemma E.1 (Slud's Inequality).** *Let $X \sim Binomial(m, p)$ with parameter $0 < p \le 1/2$. For any $0 \le m_0 \le m(1 - 2p)$,*

$$\mathbb{P}\left(X \ge mp + m_0\right) \ge \frac{1}{4} \exp\left(\frac{-m_0^2}{mp(1-p)}\right).$$

**Lemma E.2 (Chernoff Bound for Binomials).** *Let $X \sim Binomial(m, p)$, then for any $\delta > 0$,*

$$\mathbb{P}\left(X \ge (1 + \delta)mp\right) \le \exp\left(-\frac{mp\delta^2}{2 + \delta}\right).$$

*Moreover, for any $\delta \in (0, 1)$,*

$$\mathbb{P}\left(X \le (1 - \delta)mp\right) \le \exp\left(-\frac{mp\delta^2}{2}\right).$$

**Lemma E.3 (Hoeffding's Inequality).** *Let $Z_1, \ldots, Z_n$ be independent random variables in $[a, b]$ with some constants $a < b$, let $\bar{Z} := \frac{1}{n} \sum_{i=1}^{n} Z_i$. Then for all $t > 0$,*

$$\mathbb{P}\left(\left|\bar{Z} - \mathbb{E}[\bar{Z}]\right| \ge t\right) \le 2 \exp\left\{-\frac{2nt^2}{(b - a)^2}\right\}.$$

**Lemma E.4 (Stirling's Approximation).** *For any positive integer $n$,*

$$\sqrt{2\pi} n^{n+1/2} e^{-n} \le n! \le e n^{n+1/2} e^{-n}.$$

**Lemma E.5 (Log Sum Inequality).** *Let $a_1, \ldots, a_n$ and $b_1, \ldots, b_n$ be positive numbers, we have*

$$\sum_{i=1}^{n} a_i \log\left(\frac{a_i}{b_i}\right) \ge \left(\sum_{i=1}^{n} a_i\right) \log\left(\frac{\sum_{i=1}^{n} a_i}{\sum_{i=1}^{n} b_i}\right).$$

**Lemma E.6 (Fano's Inequality for Binary Hypothesis Testing).** *Let $P_0$ and $P_1$ be two possible distributions. Let $\sigma \sim \text{Unif}(\{0, 1\})$, i.e., $\mathbb{P}(\sigma = 0) = \mathbb{P}(\sigma = 1) = 1/2$. Let $Z_\sigma \sim P_\sigma$ be a random variable and $\psi : Z_\sigma \mapsto \{0, 1\}$ be a function that returns an estimate of the index $\sigma$. Then we have*

$$\inf_{\psi} \mathbb{P}\left(\psi(Z_\sigma) \ne \sigma\right) = \frac{1}{2}\left(1 - \|P_0 - P_1\|_{TV}\right),$$

*where $\|P - Q\|_{TV}$ is called the total variation distance between distributions $P$ and $Q$. Furthermore, by **Pinsker's inequality**, we have*

$$\|P - Q\|_{TV} \le \sqrt{\frac{1}{2} \min\{\mathbb{D}_{\text{KL}}(P \| Q), \mathbb{D}_{\text{KL}}(Q \| P)\}},$$

*and thus*

$$\inf_{\psi} \mathbb{P}\left(\psi(Z_\sigma) \ne \sigma\right) \ge \frac{1}{2}\left(1 - \sqrt{\frac{1}{2} \min\{\mathbb{D}_{\text{KL}}(P_0 \| P_1), \mathbb{D}_{\text{KL}}(P_1 \| P_0)\}}\right).$$

*Remark* E.7. When $\min\{\mathbb{D}_{\text{KL}}(P \| Q), \mathbb{D}_{\text{KL}}(Q \| P)\}$ is large (larger than 2 for instance), Pinsker's inequality is vacuous, we have the following **Bretagnolle–Huber's inequality** remain non-vacuous

$$\|P - Q\|_{TV} \le \sqrt{1 - \exp\left(-\min\{\mathbb{D}_{\text{KL}}(P \| Q), \mathbb{D}_{\text{KL}}(Q \| P)\}\right)},$$

and a further refined Fano's inequality for binary hypothesis testing

$$\inf_{\psi} \mathbb{P}\left(\psi(Z_\sigma) \ne \sigma\right) \ge \frac{1}{2}\left(1 - \sqrt{1 - \exp\left(-\min\{\mathbb{D}_{\text{KL}}(P \| Q), \mathbb{D}_{\text{KL}}(Q \| P)\}\right)}\right).$$

**Lemma E.8 (Berry-Esseen Inequality).** *Let $X_1, \ldots, X_n$ be a sequence of i.i.d. random variables with mean 0, variance $\sigma^2$ and $\mathbb{E}[|X|^3] = \rho < \infty$. Let $F_n$ stands for the distribution of the normalized sum $\sigma^{-1} \sqrt{n} \bar{X}_n$ and $\Phi$ stands for the standard Gaussian distribution, we have for any $n \in \mathbb{N}$,*

$$\sup_{x \in \mathbb{R}} \left|F_n(x) - \Phi(x)\right| \le \frac{3\rho}{\sigma^3 \sqrt{n}}.$$

**Lemma E.9** ([Hanneke & Kpotufe, 2022](#), **Lemma 1**). *Let $\mathcal{H}$ be a concept class. For any $n \in \mathbb{N}$ and $\delta \in (0, 1)$, define*

$$\epsilon(n, \delta) := \frac{d_{\mathcal{H}}}{n} \log\left(\frac{n}{d_{\mathcal{H}}}\right) + \frac{1}{n} \log\left(\frac{1}{\delta}\right).$$

*Let $S_n := \{(x_i, y_i)\}_{i=1}^n$ be independent (but non-identically distributed) samples. Then with probability at least $1 - \delta$, we have for any $h, h' \in \mathcal{H}$,*

$$\mathbb{E}\left[\hat{\mathcal{E}}_{S_n}(h, h')\right] \leq \hat{\mathcal{E}}_{S_n}(h, h') + C_0 \sqrt{\min\left\{\mathbb{E}\left[\hat{P}_{S_n}(h \neq h')\right], \hat{P}_{S_n}(h \neq h')\right\} \epsilon(n, \delta)} + C_0 \cdot \epsilon(n, \delta),$$

$$\frac{1}{2}\mathbb{E}\left[\hat{P}_{S_n}(h \neq h')\right] - C_0 \cdot \epsilon(n, \delta) \leq \hat{P}_{S_n}(h \neq h') \leq 2\mathbb{E}\left[\hat{P}_{S_n}(h \neq h')\right] + C_0 \cdot \epsilon(n, \delta),$$

*where $C_0 > 0$ is some universal numerical constant.*

**Lemma E.10** ([Hanneke & Kpotufe, 2022](#), **Lemma 2**). *Consider the multitask learning setting defined in Section [2](#), for any $\boldsymbol{I} \subseteq [N]$, let $n_{\boldsymbol{I}} := \sum_{t \in \boldsymbol{I}} n_t$ and $\bar{P}_{\boldsymbol{I}} := n_{\boldsymbol{I}}^{-1} \sum_{t \in \boldsymbol{I}} n_t P_t$. For any $\delta \in (0, 1)$, on the event (from Lemma [E.9](#) with $S_n = Z_{\boldsymbol{I}} := \bigcup_{t \in \boldsymbol{I}} Z_t$) with probability at least $1 - \delta$, we have for any $h \in \mathcal{H}$ satisfying*

$$\hat{\mathcal{E}}_{Z_{\boldsymbol{I}}}(h, \hat{h}_{Z_{\boldsymbol{I}}}) \leq C_0 \sqrt{\hat{P}_{Z_{\boldsymbol{I}}}(h \neq \hat{h}_{Z_{\boldsymbol{I}}})\epsilon(n_{\boldsymbol{I}}, \delta)} + C_0 \cdot \epsilon(n_{\boldsymbol{I}}, \delta),$$

*it holds that*

$$\mathcal{E}_{\bar{P}_{\boldsymbol{I}}}(h) \leq 32 C_0^2 \left(C_\beta \cdot \epsilon(n_{\boldsymbol{I}}, \delta)\right)^{1/(2-\beta)}.$$

**Lemma E.11.** *Consider the multitask learning setting defined in Section [2](#), for any $\boldsymbol{I} \subseteq [N]$, let $n_{\boldsymbol{I}} := \sum_{t \in \boldsymbol{I}} n_t$ and $\bar{P}_{\boldsymbol{I}} := n_{\boldsymbol{I}}^{-1} \sum_{t \in \boldsymbol{I}} n_t P_t$. For any $\delta \in (0, 1)$, on the event (from Lemma [E.9](#) with $S_n = Z_{\boldsymbol{I}} := \bigcup_{t \in \boldsymbol{I}} Z_t$) with probability at least $1 - \delta$, we have for any $h^*_{\bar{P}_{\boldsymbol{I}}} \in \arg\min_{h \in \mathcal{H}} \mathcal{E}_{\bar{P}_{\boldsymbol{I}}}(h)$,*

$$\hat{\mathcal{E}}_{Z_{\boldsymbol{I}}}(h^*_{\bar{P}_{\boldsymbol{I}}}, \hat{h}_{Z_{\boldsymbol{I}}}) \leq C_0 \sqrt{\hat{P}_{Z_{\boldsymbol{I}}}(h^*_{\bar{P}_{\boldsymbol{I}}} \neq \hat{h}_{Z_{\boldsymbol{I}}})\epsilon(n_{\boldsymbol{I}}, \delta)} + C_0 \cdot \epsilon(n_{\boldsymbol{I}}, \delta).$$

*Proof of Lemma [E.11](#).* For any $\boldsymbol{I} \subseteq [N]$, let $n_{\boldsymbol{I}} := \sum_{t \in \boldsymbol{I}} n_t$ and $\bar{P}_{\boldsymbol{I}} := n_{\boldsymbol{I}}^{-1} \sum_{t \in \boldsymbol{I}} n_t P_t$. On the event from Lemma [E.9](#) with $S_n = Z_{\boldsymbol{I}}$, with probability of at least $1 - \delta$ we have for any $h, h' \in \mathcal{H}$,

$$\mathcal{E}_{\bar{P}_{\boldsymbol{I}}}(h, h') \leq \hat{\mathcal{E}}_{Z_{\boldsymbol{I}}}(h, h') + C_0 \sqrt{\min\left\{\bar{P}_{\boldsymbol{I}}(h \neq h'), \hat{P}_{Z_{\boldsymbol{I}}}(h \neq h')\right\} \epsilon(n_{\boldsymbol{I}}, \delta)} + C_0 \cdot \epsilon(n_{\boldsymbol{I}}, \delta), \tag{26}$$

which follows from the fact that $\mathbb{E}[\hat{\mathcal{E}}_{Z_{\boldsymbol{I}}}(h, h')] = \mathcal{E}_{\bar{P}_{\boldsymbol{I}}}(h, h')$ and $\mathbb{E}[\hat{P}_{Z_{\boldsymbol{I}}}(h \neq h')] = \bar{P}_{\boldsymbol{I}}(h \neq h')$. Now we plug into [(26)](#) with $h = \hat{h}_{Z_{\boldsymbol{I}}}$, $h' = h^*_{\bar{P}_{\boldsymbol{I}}}$ and get

$$\mathcal{E}_{\bar{P}_{\boldsymbol{I}}}(\hat{h}_{Z_{\boldsymbol{I}}}) \leq \hat{\mathcal{E}}_{Z_{\boldsymbol{I}}}(\hat{h}_{Z_{\boldsymbol{I}}}, h^*_{\bar{P}_{\boldsymbol{I}}}) + C_0 \sqrt{\min\left\{\bar{P}_{\boldsymbol{I}}(\hat{h}_{Z_{\boldsymbol{I}}} \neq h^*_{\bar{P}_{\boldsymbol{I}}}), \hat{P}_{Z_{\boldsymbol{I}}}(\hat{h}_{Z_{\boldsymbol{I}}} \neq h^*_{\bar{P}_{\boldsymbol{I}}})\right\} \epsilon(n_{\boldsymbol{I}}, \delta)} + C_0 \cdot \epsilon(n_{\boldsymbol{I}}, \delta).$$

Note that $\hat{\mathcal{E}}_{Z_{\boldsymbol{I}}}(\hat{h}_{Z_{\boldsymbol{I}}}, h^*_{\bar{P}_{\boldsymbol{I}}}) = -\hat{\mathcal{E}}_{Z_{\boldsymbol{I}}}(h^*_{\bar{P}_{\boldsymbol{I}}}, \hat{h}_{Z_{\boldsymbol{I}}})$ and thus

$$\hat{\mathcal{E}}_{Z_{\boldsymbol{I}}}(h^*_{\bar{P}_{\boldsymbol{I}}}, \hat{h}_{Z_{\boldsymbol{I}}}) \leq C_0 \sqrt{\min\left\{\bar{P}_{\boldsymbol{I}}(\hat{h}_{Z_{\boldsymbol{I}}} \neq h^*_{\bar{P}_{\boldsymbol{I}}}), \hat{P}_{Z_{\boldsymbol{I}}}(\hat{h}_{Z_{\boldsymbol{I}}} \neq h^*_{\bar{P}_{\boldsymbol{I}}})\right\} \epsilon(n_{\boldsymbol{I}}, \delta)} + C_0 \cdot \epsilon(n_{\boldsymbol{I}}, \delta) - \mathcal{E}_{\bar{P}_{\boldsymbol{I}}}(\hat{h}_{Z_{\boldsymbol{I}}})$$

$$\leq C_0 \sqrt{\min\left\{\bar{P}_{\boldsymbol{I}}(\hat{h}_{Z_{\boldsymbol{I}}} \neq h^*_{\bar{P}_{\boldsymbol{I}}}), \hat{P}_{Z_{\boldsymbol{I}}}(\hat{h}_{Z_{\boldsymbol{I}}} \neq h^*_{\bar{P}_{\boldsymbol{I}}})\right\} \epsilon(n_{\boldsymbol{I}}, \delta)} + C_0 \cdot \epsilon(n_{\boldsymbol{I}}, \delta)$$

$$\leq C_0 \sqrt{\hat{P}_{Z_{\boldsymbol{I}}}(\hat{h}_{Z_{\boldsymbol{I}}} \neq h^*_{\bar{P}_{\boldsymbol{I}}})\epsilon(n_{\boldsymbol{I}}, \delta)} + C_0 \cdot \epsilon(n_{\boldsymbol{I}}, \delta),$$

where the second inequality follows from that the excess risk is always non-negative. $\square$

