# OpenReview forum: "When More Data Doesn't Help: Limits of Adaptation in Multitask Learning"
_ICML.cc/2026/Conference — ICML 2026 regular_

### Official Review · Reviewer_bRBr · 2026-03-06

**Soundness:** 4
**Presentation:** 3
**Significance:** 3
**Originality:** 3
**Overall Recommendation:** 4
**Confidence:** 3

**Summary:**

This paper investigates the hardness of adaptation for multitask learning. The results are built on task sets with ordered transfer exponent, which is a key quantity in the literature of transfer learning and multitask learning. The authors first point out that some impossibility results in a previous work (e.g., no-free-lunch theorem for multitask learning in (Hanneke & Kpotufe, 2022)) requires assumptions that are too strong, then provide a new construction of multitask learning problems that can relax some of the assumptions required before, but admit same no-free-lunch theorem such that any adaptive learning algorithm that only has access to source data have a minimax error rate in the constructed multitask learning problem.

**Compliance With Llm Reviewing Policy:**

Affirmed.

**Final Justification:**

My concerns regarding technical contribution has been resolved by the authors' response. I keep my score for a weak accept.

**Key Questions For Authors:**

- (Q1) Regarding the proof of Theorem 5.2, the authors say that the technique of constructing a randomized multitask learning problem "has also been adopted in Hanneke & Kpotufe (2022)". Meanwhile, the authors have mentioned before that the technical efforts in this paper is different from theirs. Since Theorem 5.2 is one of the key results in this paper, could the authors detailedly introduce that besides the construction of task sets, which key steps are also new (or remains similar) in the proof of the no-free-lunch theorem in Hanneke & Kpotufe (2022)?
- (Q2) While the assumption regarding small sample size has been relaxed, there are still two key assumptions or requirements: (a1) same optimal classifier, and (a2) large task numbers comparing to sample size. Both assumptions are still far from the algorithmic and applicative researches, where the collected data are from various sources that seem to have different individually optimal classifiers, and each dataset have a large amount of data while the number of datasets can be seen as a small constant. Can the authors discuss whether the remaining assumptions (a1) and (a2) can be further relaxed to approach more practical scenarios, or perhaps a new framework is required if we wish to have a theory that reveals more practical insights?

**Limitations:**

yes

**Strengths And Weaknesses:**

- Strengths: The paper is technically sound. The symbols and the claims in theorems are standard, the key steps of the proofs are properly introduced in main text, which makes the overall technical contribution easy to understand. The paper is generally well-written. Specifically, Section 5 is well-structured, which clearly presents the formal version of the main results in this paper. The problem investigated in this paper is definitely important, and the results in this paper push forward the study of hardness of multitask learning.

- Weaknesses: In Section 3, the authors first review a recent work which provides a no-free-lunch type theorem for multitask learning, then present the different construction in this paper. As both works are constructing "hard instances" to prove impossibility results, perhaps the most intuitive comparison that can reveal their difference is the different assumptions or properties the constructions admit. Therefore, it would be better if the authors could make such comparison clearer (e.g., elaborate on line 173-175 on the right half of page 4 to be a separate paragraph).

For other comments, please refer to "Key Questions For Authors" below.

---

> ### Author Rebuttal · Authors · 2026-03-28
>
> We sincerely thank the reviewer for taking your time and efforts in reviewing our paper. We appreciate your positive remarks regarding the contribution of this work, as well as the quality of our presentation. Below, we provide detailed responses to your questions and concerns.
>
> 1. "Therefore, it would be better if the authors could make such comparison clearer."
>
> We thank the reviewer for providing us with this insightful suggestion about elaborating more on comparing the main differences between the hard-case construction in [HK22] and ours. We will definitely make a complement if preparing for the camera-ready version. Here, we would also like to include a brief comparison. In Section 3.1, we provided an explanation on how the hard-instance in [HK22] makes adaptation impossible. Basically, their multitask construction has a few benign datasets consisting of perfectly correctly-labelled examples (called homogeneous examples). An ideal learning algorithm will achieve (nearly) minimax rates if it can pick almost all of these benign datasets out. However, their construction hides these benign datasets inside a bunch of noisy datasets. While a single noisy dataset probably contains roughly half $+1$ half $-1$ labelled examples, having enough such noisy datasets makes the probability that a few of them consist of all incorrectly-labelled homogeneous examples nontrivial (the probably can be bounded above a constant). This is where the requirement of exponentially large number of tasks ($N=\Omega(\exp(n))$) comes from. Moreover, they also have to confuse the learner about which label is correct. This is exactly done by restricting the samples size per task ($n<2/\beta-1$) such that comparing homogeneous examples cannot determine the correct label. Our probabilistic construction releases this significant constraints by removing those benign datasets, and thus excluding the option of comparing by comparing homogeneous examples.
>
> 2. "Which key steps are new in the proof compared to [HK22]?"
>
> This is a very important question we would definitely like to elaborate on. We hope that our explanation below could convince the reviewer about the strong technical novelty of our work. We will also include it in our paper if preparing for the camera-ready version. Besides having a different probabilistic construction, we clarify the following technical contributions. Most importantly, our proof strategy does differ a lot from the previous work. Our technique is to directly bound the KL-divergence between likelihood functions (of mixture distributions) when data are generated according to opposite labels. In contrast, the lower bound in [HK22] relies on the fact that likelihood-ratio test is optimal. Note that for the same problem, bounding likelihood-ratio is conceptually equivalent to bounding the KL-divergence. However, as discussed earlier, their hard-case construction is relatively strong, containing certain amount of benign tasks. And it is indeed this strong setup allows them to lower bound the likelihood-ratio by a decomposition into the contributions of homogeneous vs non-homogenous examples (see their Proposition 2). This decomposition significantly reduces the technical difficulty. (The reviewer can roughly think about this as reducing the problem from bounding a KL-divergence between mixtures to bounding a KL-divergence between joint distributions with pairwise independent marginals.) In other words, their strong construction is somehow cheating for them in technical analysis, but their result pays the price of having a significant constraint of $n<2/\beta-1$.
>
> 3. "Can the authors discuss whether the remaining assumptions can be further relaxed?"
>
> Regarding tasks sharing the same optimal classifier, we would like to emphasize that since we were proving a negative result (lower bound), the stronger the assumptions are, the stronger the negative result is. The reviewer might find it helpful to think about this from the following perspective: Even if under the very ideal assumption that all distributions share the same optimal classifier, we can still show that adaptivity is impossible. Regarding the number of tasks, since this work focuses on the minimax rates (handling the worst-case multi-source distributions), improving guarantees on $N$ probably relies on designing adaptive learning algorithms that uniformly outperform pooling. One potentially better learner might be some algorithm that adaptively checks certain aggregations of the datasets and figures out the optimal aggregation of the datasets. However, this minimax characterization is usually too pessimistic to explain practical learning performance. Hence, it would be interesting to study the rates when the data distributions are fixed, a model called universal learning [BHM$^{+}$21], where one might hope to realize adaptivity much easier (e.g. for $N=\mathrm{poly}(n)$).
>
> [HK22]: Hanneke & Kpotufe (2022)
>
> [BHM$^{+}$21]: A theory of universal learning

---

> > ### Author Rebuttal · Reviewer_bRBr · 2026-04-01
> >
> > Thanks the authors for their response, especially for their illustrative example in response to Q3 to help me see that "impossibility results under strong assumptions are strong". After reading the response to me as well as to the other reviewers, my concerns have been generally resolved. I decide to keep my score (but not increasing to 5), for the reason that **while this paper obtain some new results, the minimax characterization is usually too pessimistic to explain practical learning performance** which is also acknowledged by the authors' response. I think this restrict the practical implications from this line of work using minimax characterizations, which is a weakness of the framework, rather than the core contribution of the paper itself.

---

> > > ### Author Response · Authors · 2026-04-03
> > >
> > > We sincerely thank the reviewer for replying promptly to our rebuttal and thus leaving us enough time to (hopefully) address your remaining concerns. We are glad that our rebuttal has addressed most of your questions and concerns. We also appreciate that you clearly explain the reason for keeping the score rather than increasing it. In this reply, we still want to kindly clarify certain points.
> > >
> > > (i) We proposed this distribution-dependent variant of learning since you asked "whether a new framework is required if we wish to have a theory that reveals more practical insights". We do think that this universal framework is very interesting and we will definitely study it for future work. However, a new framework is interesting and more practical does not mean that the current fundamental problem is not worth exploring, especially when the previous work of [HK22] has left an open question of whether adaptation is possible if we get rid of the significant constraint of $n<2/\beta-1$. (Intuitively, having more samples per task might help. We also hoped that it is true but we proved it is the opposite.) Indeed, the universal learning framework of [BMH$^{+}$21] was developed after a thorough understanding of the classical PAC learning, which focuses on characterizing the distribution-free (minimax optimal) sample complexity. We believe that studying the limits of adaptation under this uniform model would benefit us to understand more practical scenarios of multitask learning.
> > >
> > > (ii) In machine learning theory, most lower bounds are proved from such a minimax perspective. Even for the universal learning model we mentioned, the lower bounds therein are developed based on analyzing the worst-case distributions as well.
> > >
> > > (iii) Our theory does directly reveal the following practical implication: For researchers/engineers who hope to design better multitask learning algorithms, they need to somehow leverage additional information rather than merely rely on data. For example, knowing which datasets are less contaminated or which sources are more relevant to the target environment would be helpful.
> > >
> > > Thanks again for taking time to read this message! We sincerely hope that you could re-evaluate our work. We would be happy to address any additional concerns.

---

### Official Review · Reviewer_qnua · 2026-03-11

**Soundness:** 3
**Presentation:** 1
**Significance:** 2
**Originality:** 2
**Overall Recommendation:** 3
**Confidence:** 2

**Summary:**

The paper provides a statistical learning theoretic impossibility result in a multitask learning context. The authors prove that *adaptivity* --- the identification of datasets arising from helpful tasks --- is not always feasible even when each dataset is large. Specifically, they provide a lower bound on error that depends both on the number of datasets and the size of each dataset.

**Compliance With Llm Reviewing Policy:**

Affirmed.

**Final Justification:**

The authors’ rebuttal and other reviews have helped clarify key aspects of the paper's significance and originality, which has resolved the concerns I raised about these dimensions of the paper. In light of this, I raised my score and lowered my confidence. I maintain my concern that many of the core ideas in the paper are not effectively conveyed to a general audience.

**Key Questions For Authors:**

I have a couple of questions regarding Algorithm 1:
1. Does Algorithm 1 leverage insights from the theoretical results? If so, which results?
2. Could you please clarify whether Algorithm 1 is a generally applicable algorithm or applicable only to the designed problem described in Section 4? How general are the conclusions from Theorem 4.4 about the difficulty of identifying relevant tasks?

In addition,

3. Could you please comment on the realisticness of, and elaborate on the sensitivity of your results to, the assumptions listed on pg. 3?

**Limitations:**

yes

**Strengths And Weaknesses:**

**Soundness:** The paper's claims are supported by a thorough theoretical analysis. I appreciate that this is a theoretical paper and the important potential contributions of purely theoretical work. I think these theoretical contributions could have been strengthened by an empirical demonstration of the authors' results in the designed problems, which I would have found helpful for developing intuition for and understanding the significance of these results.

**Presentation:** I found the technical results quite difficult to follow. I would suggest presenting the theoretical results so they are clearly delineated and self-contained (i.e., assumptions and required definitions stated or referred to in the statement of the result).

I think the relative difficulty I had following the results could also be addressed by a reorganization of the paper. It's not entirely clear to me how the results technically or conceptually relate to each other. For example, although Section 3 is titled "Main Results", the formal statement of the main results appears to be deferred to Section 5.

**Significance:** The topic of multitask learnability is practically important: Datasets encountered in practice often consist of data pooled from multiple sources. However, I have several concerns about the paper's potential significance:
- The assumptions listed on pg. 3 strike me as quite restrictive, and I'm not sure how applicable the authors' results would be in practice. In particular, assumption (1) that all data distributions induce the same optimal classifier seems like it would preclude many practical settings in which data sources are sufficiently different and/or the concept class sufficiently expressive.
- Similarly, assumption (2) on pg. 6 seems to restrict the number of tasks to be large as a function of $n$. The apparent insight that adaptivity requires the amount of data to scale with the number of tasks seems somewhat weaker than the authors' main conclusion that "adaptivity is impossible for arbitrarily large sample size per task".
- While the impossibility results are conceptually interesting, it is not clear to me what actionable insights to take from the authors' analysis.

**Originality:** The paper builds on the work of Hanneke & Kpotufe (2022), providing a result in the same setting and that does not depend on the assumption that $n$ is "small". Although the stronger impossibility result is conceptually interesting, the work strikes me as incremental: The technical departures from the work of Hanneke & Kpotufe (2022) are limited. Conceptually, the main conclusion that "adaptivity is difficult to impossible" corroborates insights from Hanneke & Kpotufe, and I have difficulty identifying the conceptual insights or practical significance the work provides, over and above those of Hanneke & Kpotufe (see also my comments regarding Significance). I would find it very helpful, and would be open to adjusting my score, if the authors could elaborate on these during the rebuttal phase.

---

> ### Author Rebuttal · Authors · 2026-03-28
>
> We sincerely thank the reviewer for taking your time and efforts to review our paper. Below, we provide detailed responses to your questions and concerns.
>
> 1. About empirical demonstration:
>
> Our work is in its nature theoretical and our main result is a no-free-lunch theorem (negative result/lower bound) on the hardness of adaptation. I don't know how to run experiments to justify it, i.e. How many algorithms we have to test in order to show that adaptivity is impossible.
>
> 2. About the assumptions on pg. 3 and pg. 6:
>
> Note that for proving a negative result, the stronger the assumptions are, the stronger the negative result is. Hence, (i) regarding the assumption that all distributions share the same optimal classifier, the reviewer might find it helpful to think in the following way: Even if under the very ideal assumption that all distributions share the same optimal classifier, we can still show that adaptivity is impossible. Adaptation is even harder and thus impossible under heterogeneous sources. (ii) Regarding Bernstein class condition, the most general setting of machine learning is the agnostic case, where no restriction is put on the label noise (corresponding to $\beta=0$). The Bernstein class condition is thus stronger than the agnostic case, making our lower bound stronger. (iii) Finally, transfer exponent has been commonly adopted as a notion to characterize the discrepancies between tasks in multitask/transfer learning. Especially for our work, since we focus on the minimax rates characterized by transfer exponents, we kept using this notion for analysis.
>
> The previous work [HK22] established a lower bound under the assumptions of $N=\Omega(\exp(n))$ and $n<2/\beta-1$. We get rid of the latter in this paper. Our main result is, for arbitrarily large $n$, there exists a multitask problem (distributions and classes) such that adaptivity is impossible. The hard-instance construction for proving a no-free-lunch theorem always relies on scaling the distributions with sample size $n$ (or equivalently $1/\epsilon$) The reviewer might be interested in looking at the classical no-free-lunch theorem for PAC learning.
>
> 3. "The technical departures from the work of [HK22] are limited."
>
> This is a very important question we would definitely like to elaborate on. We hope that our explanation below could convince the reviewer about the strong technical novelty of our work. Besides having a different probabilistic construction, we clarify the following technical contributions. Most importantly, our proof strategy does differ a lot from the previous work. Our technique is to directly bound the KL-divergence between likelihood functions (of mixture distributions) when data are generated according to opposite labels. In contrast, the lower bound in [HK22] relies on the fact that likelihood-ratio test is optimal. Note that for the same problem, bounding likelihood-ratio is conceptually equivalent to bounding the KL-divergence. However, as discussed in the paper, their hard-case construction is relatively strong, containing certain benign tasks. And it is indeed this strong setup allows them to lower bound the likelihood-ratio by a decomposition into the contributions of homogeneous vs non-homogenous examples (see their Proposition 2). This decomposition significantly reduces the technical difficulty. (You can roughly think about this as reducing the problem from bounding a KL-divergence between mixtures to bounding a KL-divergence between joint distributions with pairwise independent marginals.) In other words, their strong construction is somehow cheating for them in technical analysis, but their result pays the price of having a significant constraint of $n<2/\beta-1$.
>
> 4. About Algorithm 1:
>
> Firstly, we would like to emphasize that Algorithm 1 is not the main contribution of our work, its only purpose is to prove that pooling is suboptimal sometimes (Algorithm 1 yields a better upper bound than pooling.). Now, to answer your questions, Algorithm 1 is always applicable to any multitask learning problem given multi-source datasets and a set of transfer exponents. But it is not always optimal adaptive. Theorem 4.4 basically states that: (i) For the specific multitask problem constructed in Section 4, Algorithm 1 not only outperforms pooling, but also admits learning rates faster than the minimax rates (which accounts for the worst-case scenarios). (ii) Algorithm 1 guarantees nearly optimal uniform rates (differing only by a $\log(N)$ factor to the minimax rates) given a certain set of transfer exponents $(1,\infty,\ldots,\infty)$. Note that the multitask distributions constructed in Section 4 also admits $(1,\infty,\ldots,\infty)$ transfer exponents. The authors do believe a clever designed variant of Algorithm 1 could be a potentially better adaptive learner. Finally, Theorem 4.4 is an upper bound and does not relate to the difficulty of learning.
>
> [HK22]: A no-free-lunch theorem for multitask learning.

---

> > ### Author Rebuttal · Reviewer_qnua · 2026-04-01
> >
> > Thank you to the authors for the detailed responses to my concerns and questions. After reading the authors’ rebuttal and other reviews, I better appreciate both the significance of the work’s departure from Hanneke & Kpotufe (2022), and the role of the authors’ assumptions. I will raise my score in light of my now higher assessment of the paper’s significance and originality. In part due to the presentation of the paper, I did not understand central parts of the paper and its contributions which have been raised in the rebuttal and other reviews. Because of this, I will also lower my confidence.
> >
> > I maintain my concern that many of the core ideas in the paper, including core aspects of the significance of the paper’s technical contributions, are not effectively conveyed to a general audience. I would make the following suggestions to the authors:
> > - As mentioned in my initial review, I find the result that adaptivity requires the amount of data to scale with the number of tasks to be weaker than the authors’ conclusion that “adaptivity is impossible for arbitrarily large sample size per task”. I suggest emphasizing the requirement that $N$ scales with $n$: How does this affect the interpretation of the main results? Would it be interesting or feasible to remove this dependence? Some discussion of this is given in Section 6; it might be useful to contextualize or preview this discussion earlier in the paper.
> > - I found the current organization of the paper difficult to follow. As also suggested by Reviewer Yeqe, it may improve the flow of the paper to rearrange the order of, and include stronger motivation for each of, Sections 3–5.
> > - I suggest including a more extensive description of and motivation for Algorithm 1, perhaps in a separate subsection to highlight differences in the nature of the contributions of Theorems 4.1 and 4.2 vs. Theorems 4.3 and 4.4.

---

> > > ### Author Response · Authors · 2026-04-03
> > >
> > > We sincerely thank the reviewer for re-evaluating the significance and originality of our work, and raising your score. We appreciate that you gave us those practical suggestions on improving the presentation of the paper, and would like to apologize that our rebuttal did not address your concerns about the presentation. We thank the reviewer for replying promptly to our rebuttal. In this reply, we want to kindly reply to (and hopefully address) your remaining concerns.
> > >
> > > (i) Our theory does directly reveal the following practical implication: For researchers/engineers who hope to design better multitask learning algorithms, they need to somehow leverage additional information rather than merely rely on data. For example, knowing which datasets are less contaminated or which sources are more relevant to the target environment would be helpful. Please see also our response to the reviewer bRBr.
> > >
> > > (ii) Thank you for pointing out this non-rigorous part of our claim. Our lower bound does require $N=\Omega(\exp(n))$. There are two constraints in the previous work of [HK22], one is the absolute bound $n<2/\beta-1$ and the other is $N=\Omega(\exp(n))$. [Hk22] left an open question of whether adaptivity is possible without any (or both) of these two constraints. We hoped to develop adaptive learning algorithms that can achieve minimax rates, but turned out to be able to prove a negative result for an arbitrarily large $n\in\mathbb{N}$.
> > >
> > > Regarding your questions, A very rigorous interpretation to our main results is: For an arbitrarily large sample size per task $n$, when the number of tasks $N$ is sufficiently large, adaptation is impossible for multitask learning. However, since adaptation is defined to be that there exists a multitask learning algorithm that, having only access to the data, can achieve the asymptotic minimax rates (asymptotic with respect to both $n$ and $N$). We think it is still reasonable to simply say that adaptation is impossible with unlimited samples per task. We will clarify this in the camera-ready version.
> > >
> > > Moreover, proving a lower bound without the constraint of $N=\Omega(\exp(n))$ might be possible but definitely not interesting. It would be more interesting if the opposite happens, i.e. we can develop certain adaptive algorithms (to achieve the minimax rates) when, for example, only having $N=O(\mathrm{poly}(n))$ tasks. As we discussed in the paper, this might require us to figure out the answer to the optimal adaptivity first. We leave it for future work.
> > >
> > > (ii) We will definitely refine the organization of the paper, explain more about the motivation for each section in Sections 3-5, and highlight the technical novelties (contribution and departure from the previous work) of the theorems in Section 4 and 5, for the camera-ready version.
> > >
> > > Thank you for taking time to read this message! We really hope that you could re-evaluate the quality of our work. Please let us know if you have further concerns.

---

### Official Review · Reviewer_UufA · 2026-03-13

**Soundness:** 3
**Presentation:** 2
**Significance:** 3
**Originality:** 3
**Overall Recommendation:** 4
**Confidence:** 3

**Summary:**

This paper studies the impossibility of adaptation in multitask learning from a statistical learning theory perspective. The central question is whether having a larger sample size per task could enable adaptive algorithms to achieve minimax rates. The authors answer negatively: they construct a multitask problem where, for an arbitrarily large sample size per task n, no adaptive algorithm can achieve the minimax rate, provided the number of tasks N is super-exponentially large in n. The proof leverages Fano's method combined with a tight KL divergence bound between mixture distributions. Another contribution, the paper demonstrates that pooling (global ERM) is suboptimal in the agnostic case but achieves nearly optimal adaptive rates in the main construction.

**Compliance With Llm Reviewing Policy:**

Affirmed.

**Final Justification:**

The author has addressed my concerns, so I will keep my score.

**Key Questions For Authors:**

1. The pooling optimality claim (Section 5.3) — does it extend beyond the specific construction, or is it purely a property of the designed problem?

2. Can the authors provide a concrete example satisfying all conditions of Theorem D.1 to illustrate its significance?

3. Is there any evidence or conjecture on what the optimal adaptive rates are, beyond showing pooling is sometimes nearly optimal?

**Limitations:**

Yes

**Strengths And Weaknesses:**

## Strengths

- The paper directly resolves an open question from Hanneke & Kpotufe (2022), providing a stronger impossibility result that holds for arbitrarily large sample size per task — a meaningful and well-motivated theoretical contribution.

- The conceptual message is important and clear: abundant data per task cannot overcome the fundamental hardness of adaptation when the number of tasks is large.
- The paper is well-organized, situates itself carefully relative to prior work, and covers the related literature thoroughly.


## Weaknesses

- The claim that "pooling achieves nearly optimal adaptive rates" (Section 5.3) holds only within the specific constructed multitask problem, not in general multitask settings. This should be stated more carefully.
- The construction is restricted to binary classification over a two-point instance space $\{x_0, x_1\}$ with a very specific distributional family, which limits the generality of the conclusions.
- The paper frames optimal adaptivity as future work but provides no new upper bound on optimal adaptive rates, leaving the gap between what is achievable and what is impossible entirely uncharacterized.

## Ref:

Hanneke & Kpotufe (2022), A No-Free-Lunch Theorem for MultiTask Learning

---

> ### Author Rebuttal · Authors · 2026-03-27
>
> We sincerely thank the reviewer for your efforts in taking the time to read and review our paper. We appreciate your positive remarks regarding the theoretical contribution and significance of this work, as well as the quality of our presentation. Below, we provide detailed responses to your questions and concerns.
>
> 1. About the claim that "pooling achieves nearly optimal adaptive rates":
>
> We thank the reviewer for pointing out this issue. This is definitely not a rigorous section title and we will address the issue if preparing for the camera-ready version.
>
> 2. The construction is restricted to binary classification with a very specific distributional family:
>
> Regarding the two-points distribution, we would like to emphasize that since we were proving a negative result (lower bound), the stronger the assumptions are, the stronger the negative result is. The reviewer might find it helpful to think about this from the following perspective: Even if under the very ideal/strong assumptions on the distribution, we can still show that adaptivity is impossible. Moreover, we would like to emphasize that our hard-case example (as well as the one in [HK22]) even allows the learner to know the distributions, i.e., for any adaptive learner, as long as having no access to the information which datasets are drawn from the fair distribution, it cannot achieve minimax optimal rates even if it knows the exact construction of the fair and noisy distributions. In conclusion, we believe this simple two-points distribution makes our lower bound very strong. Regarding the binary classification setting, since the main focus of our work is to release a significant constraint on the sample size per task ($n<2/\beta-1$) required in the previous work of [HK22], we studied the same binary classification problem as they did. Generally speaking, learning theory researches usually start with the binary classification setup and then extend to other general settings. Definitely, it would be very interesting to study adaptation of multitask learning under other learning settings such as multi-class classification, regression and bandits problems.
>
> 3. The paper leaves the optimal adaptivity entirely uncharacterized:
>
> We thank the reviewer for this insightful suggestion. Providing results on optimal adaptive rates would definitely enhance the quality of the work. Unfortunately, we found that understanding general optimal adaptivity is very challenging, even if finding an adaptive multitask learning algorithm that uniformly outperform pooling is difficult. As we discussed in the paper, characterizing the minimax optimal adaptive rates might rely on developing tight bounds on the KL divergence between mixture distributions. To the best of our knowledge, such type of theory is lagging behind. Hence, we left it as an open question for future works.
>
> 4. For the key questions about optimal adaptivity:
>
> We thank the reviewer for proposing these very insightful questions. Honestly, we found that pooling achieves optimal adaptive rates for our constructed learning problem coincidently, and then tried to extend the result in Theorem D.1. Since it is known that pooling is optimal when a constant quantile of tasks are good (Lemma 5.3), it would be interesting to study the case when the number of good tasks is small. Theorem D.1 basically implies the following: when only a few tasks are good, pooling achieves nearly optimal adaptive rates when bad tasks are pretty bad. The main intuition behind is to hide a few good tasks into a large set of bad tasks so that no adaptive learner can figure out where are them, and a random aggregation (probably full) of bad tasks is even less efficient than pooling. A simple example that satisfies the conditions in Theorem D.1 is exactly our construction in Section 5. We re-formulate our theory as presented in Theorem D.1 because, ideally we would hope to provide a clean characterization based on the set of transfer exponents to identify whether pooling is optimal among adaptive algorithms. We would definite elaborate more on our Theorem D.1 if preparing for the camera-ready version. Finally, we would like to point out again that finding an adaptive multitask learning algorithm that uniformly outperforms pooling is difficult. By saying "uniformly outperform", we refer to the worst-case multi-source distributions that admit a given set of transfer exponents. We conjecture that one candidate might be some algorithm that adaptively checks certain aggregations of the datasets (note that checking all possible aggregations of $N$ datasets will result in $2^{N}$ combinations and will fail the union bound) and roughly figure out the optimal aggregation of the datasets.
>
> References:
>
> [HK22]: Hanneke & Kpotufe. A no-free-lunch theorem for multitask learning. The Annals of Statistics.

---

> > ### Author Rebuttal · Reviewer_UufA · 2026-04-04
> >
> > Thank you for the detailed rebuttal. My concerns have been addressed. I will keep my score.

---

> > > ### Author Response · Authors · 2026-04-06
> > >
> > > Thank you for carefully reading our rebuttal and for your encouraging feedback. We are happy to know that you found our response clear and helpful. Your insightful comments will definitely help us improve the paper.

---

### Official Review · Reviewer_Yeqe · 2026-03-16

**Soundness:** 3
**Presentation:** 2
**Significance:** 3
**Originality:** 3
**Overall Recommendation:** 4
**Confidence:** 3

**Summary:**

This paper studies the statistical limits of adaptation in multitask learning, where the goal is to use datasets from multiple source distributions to improve prediction on a target distribution. Here, adaptivity refers to an algorithm's ability to automatically identify from the datasets alone which sources are helpful, without any prior distributional information. Prior work by Hanneke and Kpotufe (2022) has shown a no-free-lunch result: using a bounded number of samples per source task, no adaptive algorithm can achieve the minimax rates.

This paper builds on the result and provides a stronger negative result. It introduces a new construction which shows that adaptivity is impossible even with an arbitrarily large per-task sample size, given that the number of source tasks is large. The paper also shows that, for this construction, simply pooling all datasets together is near-optimal among all adaptive algorithms.

**Compliance With Llm Reviewing Policy:**

Affirmed.

**Key Questions For Authors:**

- The paper focuses on binary classification. Do you expect the results to extend to other settings such as regression? Would this require a very different set of technical tools?
- Do you have a conjecture about what happens when the number of tasks is $o(\exp(n))$?
- Beyond the two specific constructions in the paper, is there a more general characterization or intuition for when pooling should be expected to be a near optimal adaptive algorithm?
- The notion of transfer exponent can feel a bit abstract. How well does it capture transfer or multitask learning in practice?

**Limitations:**

yes

**Strengths And Weaknesses:**

*Soundness*:

- The paper appears technically sound overall. I did not check the proofs in the appendix but the construction and the proof technique seem reasonable and appropriate.

*Presentation*:

- Overall, the paper is well written and reasonably easy to follow given the technical nature of the material. That said, the exposition is fairly dense and can be a little dry in places.
- I found the placement of Section 4 to be a little awkward. Section 3.2 seems to lead quite naturally into Section 5. While Section 4 provides useful intuition, its role in the overall narrative was not immediately clear on my first reading. Particularly, as the authors point out, studying the agnostic case does not prove or disprove whether the hardness can be overcome with more data per task.
- One minor thing: the notion of a semi-adaptive learner in Section 3.1 does not seem to be defined.

*Significance*:

- It is well known in multitask learning that naively aggregating data from heterogeneous sources may not be helpful and can even lead to negative transfer, but it is less understood when adaptation is fundamentally impossible. In that sense, the paper's sharper characterization of the limits of adaptation is a valuable contribution. Even though the main result is somewhat pessimistic, it is still of interest to the community, both for improving the theoretical understanding of multitask learning and for motivating several interesting open questions.
- That said, the main result requires the number of source tasks to be exponentially large, so it still leaves open the question of whether adaptivity is possible in regimes with a more moderate number of tasks.
- The results are formulated through the notion of transfer exponent. While this seems a clean way to quantify how useful each source is in theory, it is less clear to me how it captures the kinds of structure in practical multitask or transfer learning settings. A few concrete examples would be helpful.
- The paper also relies on the assumption that all distributions share the same optimal classifier. Relaxing this to a shared representation would help make the results more broadly applicable.

*Originality*:

- The paper builds on prior work and introduces a new construction that leads to a stronger impossibility result which removes the sample size constraint. It also provides new insights on the optimality of the pooling algorithm. The proof technique appears to be built on (Hanneke & Kpotufe, 2022).

---

> ### Author Rebuttal · Authors · 2026-03-27
>
> We sincerely thank the reviewer for your time to read and review our paper. We appreciate your positive remarks regarding the contribution and significance of this work, as well as the quality of our presentation. We first respond to your specific concern about the assumption of the same optimal classifier among tasks. For proving a negative result (lower bound), the stronger the assumptions are, the stronger the negative result is. You may find it helpful to think as: even if under the very ideal assumption that all distributions share the same optimal classifier, we can still show that adaptation is impossible.
>
> Next, we provide responses to your remaining questions and concerns.
> 1. The awkward placement of Section 4:
>
> We agree that Section 4 is a bit out of place. Section 4 is supposed to provide a toy example to convince the readers that adaptivity is impossible even with unlimited data per task. We hoped to provide some intuitions on why the construction in Section 3.2 yields a stronger lower bound. We chose the agnostic case because: (i) The agnostic case ($\beta=0$) puts no assumption on the label noise and is more general than the Bernstein class condition. (ii) The agnostic case is better known to the community and its analysis is much easier. (iii) While our agnostic lower bound does not guarantee whether we can remove the constraint $n<2/\beta-1$ in the work of [HK22], it does allow unlimited data per task, thus shedding light on proving such a lower bound under the stronger Bernstein class condition. We will definitely address this confusion if preparing for the camera-ready version.
>
> 2. Missing definition of the semi-adaptive learner:
>
> We thank the reviewer for pointing out this issue. It is a learner proposed by [HK22] that has access to the ranking information of the transfer exponents. We should definitely have included a brief explanation since we mentioned it. We will address it later.
>
> 3. About transfer exponent:
>
> This notion of transfer exponent is one of the discrepancy characterizations for classification problem under transfer learning. It has been proposed and studied by previous theoretical works on multitask/transfer learning, e.g. [HK19, KK24, HK22, HK24]. In those works, the authors have included comprehensive discussions about the motivation of transfer exponent, other discrepancy characterizations, as well as certain concrete examples. We kept using this notion because our work focuses on the minimax rates characterized by transfer exponents. We omitted this discussion due to the space limitation, but would definitely make a complement later.
>
> 4. Extension to other settings such as regression:
>
> As discussed, the stronger the assumptions are, the stronger the negative result is. Hence, adaptation should also be impossible for multitask meta learning. Furthermore, we would like to share that the same technique can be applied to show that adaptation is impossible under covariates shift, where only marginal distribution changes. For a regression setting, we conjecture that adaptation is probably still impossible. However, some other notions should be formally proposed to capture the discrepancies between tasks, and thus might leading to a different construction on the hard-case example. However, proving a tight information-theoretic lower bound is usually via the well-known Assouad/Fano's method.
>
> 5. Conjecture about what happens when the number of tasks is $o(\exp(n))$:
>
> At this moment, the best-known guarantee to us is that pooling is adaptive when $N=O(\mathrm{polylog}(n))$, and the best-known lower bound still requires $N=\Omega(\exp(n))$. We believe that the correct route to resolve this large gap is to develop better multitask learners to sharpen the upper bound rather than to prove a stronger lower bound. One candidate might be some algorithm that adaptively checks certain aggregations of the datasets and figures out the optimal aggregation of the datasets.
>
> 6. Intuition for when pooling could be a nearly optimal adaptive algorithm:
>
> We thank the reviewer for proposing this insightful question. This question is different from asking when pooling is adaptive (the latter implies the former but not vice versa). Since we know pooling is optimal when a constant quantile of tasks are good (Lemma 5.3), it would be interesting to study the case when the number of good tasks is small. Our Theorem D.1 implies the following intuition: when only a few tasks are good, pooling achieves nearly optimal adaptive rates when bad tasks are pretty bad. However, understanding optimal adaptivity in multitask learning is very challenging, and we are lack of intuition on when pooling can optimally adaptive in general.
>
> [HK19]: On the value of target data in transfer learning. NeurIPS.
>
> [HK22]: A no-free-lunch theorem for multitask learning. Annals of Statistics.
>
> [HK24]: Adaptive Sample Aggregation In Transfer Learning.
>
> [KK24]: Tight Rates in Supervised Outlier Transfer Learning. ICLR.

---

> > ### Author Rebuttal · Reviewer_Yeqe · 2026-04-04
> >
> > I thank the authors for their responses. The clarifications are helpful, and my overall assessment, especially regarding the significance of the results, remains (slightly) positive. I am keeping my score.

---

> > > ### Author Response · Authors · 2026-04-06
> > >
> > > Thank you for carefully reading our rebuttal and for your encouraging feedback. We are grateful that you found our response helpful. We appreciate your insightful comments and constructive questions throughout this review process.

---

### Decision · Program_Chairs · 2026-04-30

**Decision:**

Accept (regular)

**Comment:**

The paper shows an important fundamental limitation in multitask learning settings even with large amounts of data. The presented results account for specific scenarios and regimes, but the contribution to the state-of-the-art is significant. The reviewers have also pointed out that the paper can significantly benefit from an improved presentation that makes the paper's results more easily accessible. The authors have agreed to carry out such improvements for the camera ready version, and I encourage them to do it diligently since they can really improve the paper's quality.